# Energy-Efficient Retrofit Measures (EERM) in Residential Buildings: An Application of Discrete Choice Modelling

**Clara Camarasa** [1,*] **, Lokesh Kumar Kalahasthi** [2] **, Ivan Sanchez-Díaz** [2] **, Leonardo Rosado** [2] **, Lena Hennes** [3] **, Katrin Bienge** [3] **and Ian Hamilton** [4]

1   Department of Technology, Management and Economics, UNEP-DTU Partnership, 2100 Copenhagen, Denmark
2   Department of Technology Management and Economics, Chalmers University of Technology, 41296 Gothenburg, Sweden; klokesh@chalmers.se (L.K.K.); ivan.sanchez@chalmers.se (I.S.-D.); rosado@chalmers.se (L.R.)
3   Sustainable Production and Consumption, Wuppertal Institute, 42103 Wuppertal, Germany; lena.hennes@wupperinst.org (L.H.); katrin.bienge@wupperinst.org (K.B.)
4   Energy Institute, University College London (UCL), London WC1H 0NN, UK; i.hamilton@ucl.ac.uk
*   Correspondence: clacam@dtu.dk

**Abstract:** Cross-country evidence on the adoption of energy-efficient retrofit measures (EERMs) in residential buildings is critical to supporting the development of national and pan-European policies aimed at fostering the energy performance upgrade of the building stock. In this light, the aim of this paper is to advance in the understanding of the probability of certain EERMs taking place in eight EU countries, according to a set of parameters, such as building typology, project types, and motivation behind the project. Using these parameters collected via a multi-country online survey, a set of discrete-choice (conditional logit) models are estimated on the probability of selecting a choice of any combination of 33 EERMs across the sampled countries. Results show that actions related to the building envelope are the most often-addressed across countries and single building elements or technology measures have a higher probability of being implemented. The modelling framework developed in this study contributes to the scientific community in three ways: (1) establishing an empirical relationship among EERMs and project (i.e., retrofit and deep retrofit), (2) identifying commonalities and differences across the selected countries, and (3) quantifying the probabilities and market shares of various EERMs.

**Keywords:** energy efficiency; technology diffusion; adoption; residential building stock; empirical evidence; energy conservation; Europe





## 1. Introduction

In Europe, buildings are responsible for approximately 40% of energy consumption and 36% $CO_2$ emissions [1]. Furthermore, about 35% of the building stock is over 50 years old and more than 75% is considered to be energy inefficient [2]. In view of this situation, the EU has determined to improve the energy performance of the building stock and has set out two main directives to support this task: (1) the Energy Efficiency Directive (2012/27/EU) (EED) and (2) the Energy Performance of Buildings Directive (revised 2018/844/EU) [3,4]. The EED establishes a set of binding measures to help the EU reach its 20% energy efficiency target by 2020, while the revised EPBD demands all buildings -including residential- to be nearly zero-energy (nZEB) by 2050 [5]. Defining nZEB as "a building that has a very high energy performance, as determined in accordance with Annex I. The nearly zero or very low amount of energy required should be covered to a very significant extent by energy from renewable sources, including energy from renewable sources produced on-site or nearby".

The necessary technology options to decrease building's energy demand to nZEB standards are readily available and, in many cases, economically viable [6–9]. The promising

performance and economic potential of these technologies have also been acknowledged in residential buildings at an EU level. A study developed by Hermelink et al. showed that in various cases, and depending on the exact national nZEB definition, it can even be more cost effective than other non-energy efficient solutions [10]. Nevertheless, refurbishment rates still range between 0.4 and 1.2% [11], out of which less than 5% is estimated to fulfil a high energy efficiency standard [12].

The divergence between the techno-economic potential and actual market behaviour has been coined as the 'energy efficiency gap' or 'energy paradox' and it implies that market hurdles are preventing the large-scale diffusion of these solutions [13]. To bridge the energy efficiency gap and favour the low-carbon transformation of the residential building stock, policy measures and technology selection need to be further developed favouring the uptake of energy efficient (EE) solutions. These instruments should be designed addressing the existing market drivers and barriers currently impeding the deployment of these solutions. Given the diverse national organization that composes the EU, it is particularly important to identify country-specific differences in energy-saving technology adoption patterns to generate an appropriate combination of common and country-specific policies [14,15]. By knowing how and why EE technologies are uptaken within a country, it is possible to accelerate the technology adoption process through more effectively designed programs, demonstration projects, channels of distribution, marketing strategies, and/or policy incentives [16]. Nevertheless, due to the lack of monitoring of past and present retrofit projects in the EU, this information is presently unavailable, particularly across countries [17].

Against this background, the aim of this study is to advance in the understanding of EE technology diffusion gradients in residential buildings across EU countries. Particularly, it seeks to answer the following research questions:

- What is the impact of different drivers (i.e., economic, environmental, technical, legal, and social) on the probability that certain measures or combinations of measures have been adopted in refurbishment and construction projects in the EU?
- How does this probability of technology adoption differ across building envelope measures, heating and cooling systems, and appliances?

To address these research questions, we develop a survey instrument and conduct a statistical analysis in the form of a modelling framework to estimate the probability/share of a certain choice for a given choice-set (any combination of EERMs).

## 2. Theoretical Framework

Following, we synthesize the state of research in this topic. We then elaborate on the methods used to collect and analyse the data (Section 2). Results of the conditional logit model and numerical analysis are presented in Section 3. Discussions in Section 4 followed by conclusions in Section 5. The paper closes with outlook on future research and potential applications (Section 6).

Scientific literature on residential energy use in the EU focuses mostly on a single country or region due to the profound heterogeneity in the socioeconomic contexts across EU countries, and the major challenge of collecting building stock data [18]. In the UK, for instance, Hamilton et al. investigate the combination of measures that have been installed, according to key neighbourhoods and found that the take up was higher among low-income housing that reflected government support but that proportional energy savings were greater among high-income households [19,20]. For its part, in the Netherlands, W. Poortinga et al. studied values and environmental concerns and included an extensive list of technological measures and behavioural practices associated with household energy use. The results suggest that using only attitudinal variables, such as values, may be too limited to explain all types of environmental behaviour [21]. In Germany, Michelsen et al. explored the potential advantages of housing company's size, i.e., economies of scale, economies of scope and institutional learning in thermal upgrades of residential housing, demonstrating that that large housing companies outperform private landlords by far in

high effort refurbishment projects. In contrast, private landlords appear to have advantages in low effort, incremental refurbishment activities [22]. Likewise, Achtnicht studied factors influencing German house owners' preferences on energy retrofits based on a survey of more than 400 owner-occupiers of single-family detached, semidetached, and row houses, which resulted in house owners for whom there is a favourable opportunity are more likely to undertake energy retrofit activities [23]. Kesternich derived the factors that increase the WTP for energy efficiency in the case of an upcoming move and found out that the WTP is not mainly determined by socioeconomic attributes like household income or formal education, but rather by environmental concerns and energy awareness [24]. As a consolidated tool to guide choices at the level of the single building and national policy level, Moschetti et al. combined life-cycle environmental and economic assessments in building energy renovation project of a single-family house in Norway. The results demonstrated the close to negative linear regression between the environmental and economic indicators that were computed [25]. Michelsen et al. explored motivational factors influencing the homeowners' decisions between residential heating systems in Germany and were able to demonstrate that adoption motivations can be grouped around six dimensions: (1) cost aspects, (2) general attitude towards the RHS, (3) government grant, (4) reactions to external threats (i.e., environmental or energy supply security considerations), (5) comfort considerations, and (6) influence of peers [26]. In Sweden, Nair et al. analysed the uptake of heating systems identifying that the government investment subsidy was important for conversion from a resistance heater, but not from an oil boiler. They considered several thermal energy investments as behavioural practices related to electricity and thermal energy use for owners of detached houses, concluding that personal attributes such as income, education, age, and contextual factors, including age of the house, thermal discomfort, past investment, and perceived energy cost, influence homeowners' preference for a particular type of energy efficiency measure [27]. Ek et al. focused on the role of information in energy conservation behaviour among Swedish households. The results indicate that both costs and attitudes—in particular environmental motives—are important [28].

Given that the non-existent or unavailable data in this field, most studies are supported by preliminary data collection through interviews and surveyed data. Nevertheless, comparing the findings across countries is not always possible due to the diversity of scopes, variables, and approaches that each of them has adopted. This lack of cross-country comparability, in turn, hinders a consistent pan-European overview of this field. Some of the few cross-country analyses of household adoption of energy efficient technologies on a pan-European scale are: Schleich et al. focusing energy efficient technology adoption in low-income households. They explored the diffusion of low-energy houses in France, Germany, Italy, Poland, Romania, Spain, Sweden, and the United Kingdom and discovered that homeowners falling into the lowest income quartile exhibit lower adoption propensities than those falling into the highest income quartile [29,30]. Likewise, through the REMODECE project, Almeida et al. conducted a survey in 11 EU countries addressing drivers and barriers for household and evaluating how much electricity could be saved by the use of the most efficient household appliances. Findings suggested that electronic loads and entertainment are key contributors to the power demand contributing to 72% of the total energy consumption in the standby mode [18]. Similarly, Ameli et al. examined the determinants of households' investment in energy efficiency and renewable across OECD countries, including EU representatives (i.e., France, Netherlands, Spain, and Sweden). Findings showed that homeowners and high-income households are more likely to invest than renters and low-income households. In addition, environmental attitudes and beliefs, as manifested in energy conservation practices or membership in an environmental non-governmental organisation, also play a relevant role in technology adoption [31,32].

In this way, the literature shows that the adoption, or lack thereof, of energy efficiency measures is part of a complex system of motivations and values, decisions and negotiations, interactions and experiences, and practices and beliefs [33]. There is, therefore, the need to better understand what EERMs stakeholders perceive as having the highest priority and

potential within the market they operate. Currently, there is a lack of research about what EERMs have been adopted in realized projects, how this differs across EU countries, and the motivations behind these interventions. Based on this knowledge gap and research need, this paper develops a survey to collect data on EERMs in realized projects. It then investigates the probability of a given choice of EERMs adopted in a given country, building typology and project type, based on what had been identified as most promising actions to achieve carbon targets. The work highlights the differences and similarities across EU Member States and project types (i.e., new building and retrofit projects).

## 3. Research Methodology

To describe the probabilities of various combination of EERMs being adopted across the EU for various building typologies, project types, motivations, and countries, data collected through a multi-country online survey was used to estimate discrete-choice models of the probability of technology selection, which are described in further detail in the remainder of this section.

### 3.1. Data Collection: A Multicountry Online Survey

The multi country survey developed in this study was distributed in 2019 across eight European countries: Spain (ES), Italy (IT), Poland (PL), Germany (DE), Netherlands (NL), United Kingdom (UK), France (FR), and Belgium (BE).

To test the validity of the survey design and examine any potential flaws in the research conception, a workshop was organized gathering stakeholder representatives from all relevant groups in the building value chain (i.e., supply-side actors, demand-side actors and enablers). More than 20 participants attended the workshop. The input received validated the survey design and served as a valuable basis for drafting the questionnaire.

The final questionnaire was then translated into the language and jargon of each country. The translations were revised by market experts in each country to ensure the correct understanding and interpretation of the questions within their context.

The overall survey sample contained 7231 responses [34]. The population of interest (i.e., residential building projects in the EU) was significantly large and heterogeneous, and there was a need to represent even the smallest subgroups of the population (e.g., stakeholder groups and comprehensive refurbishment projects). A stratified random sample approach was hence considered as the most appropriate. The sample was divided into three stratification axes to describe the universe (I) building typology (Table 1), (II) project type (Table 2), and (III) stakeholder groups involved in the implementation of EERMs.

**Table 1.** Characterization of building typologies and project types: definitions and acronyms.

| Cluster | Types | Description |
|---------|-------|-------------|
| Building typology | Single-dwelling building (SDBs) | Single-family house or detached house, Semi-detached (also known as twin house or duplex) Row house (also known as terraced house). |
| | Multidwelling building (MDBs) | Small multidwelling home or small apartment building, large multidwelling home or large apartment building |

**Table 2.** Characterization of project types: definitions and acronyms.

| Project Types | Description |
|---|---|
| Retrofit (including overhaul or potential repair) (R) | Upgrade the function of one or multiple building components. This can also include any necessary action to restore any broken, damaged or failed device, equipment, part or property to an acceptable usable state. |
| Deep retrofit (D) | Extra measures with the aim to upgrade the building to a higher standard. In the case of this study it refers to the inclusion of energy efficiency measures, such as insulation for the walls, ground floor and attic, air-tightening of the building's envelope, new energy efficient windows, heat recovery, solar cells, etc. |

In the survey, one stratification axis is controlled ex-ante. Namely, it was decided to what stakeholder group the survey was sent to. The other two axes were controlled ex-post. Meaning, while responses were being collected, an assessment of type of building typologies and project types that were being selected was evaluated to ensure that there were enough responses for the building typology and project type.

When the two main building typology groups and project types are combined, the following groups arise, herein referred to as "building combos":

- Retrofit (R) of single-dwelling buildings (SDBs);
- Retrofit (R) of multi-dwelling building (MDBs);
- Deep retrofit (D) of single-dwelling buildings (SDBs);
- Deep retrofit (D) of multi-dwelling buildings (MDBs).

To ensure the analysis of each axis, a minimum quota was assigned to each one. The minimum quota per country was established in 500 responses, with a correction factor in those countries with a smaller population size (e.g., Belgium). The minimum quota was defined based on an equally distributed minimum number of responses requires in each stratum (Annex I. Table A1). To obtain a well-balanced sample, the distribution of the survey was assigned to a professional company with a pan-European presence, which was able to follow a consistent methodology across all countries. The distribution collected a random sample. Since the analysis focused on building projects, a subset of the complete database was used, excluding any response that did not contain a project type. Subsequently, the final sample used for this study consists of 4277 responses, with the following distribution across countries: Spain ($n = 511$), Italy ($n = 722$), Poland ($n = 545$), Germany ($n = 526$), Netherlands ($n = 505$), United Kingdom ($n = 504$), France ($n = 501$), and Belgium ($n = 463$).

Content and Structure of the Survey

The questionnaire layout was drafted and reviewed by market experts and pretested by stakeholder representatives across the sampled countries. The main purpose of the pre-test was to ensure the comprehensibility of the questions across different contexts and levels of knowledge. It also helped to identify inconsistencies, coding errors, and data gaps. The final inquiry is composed of five sections (1–5), each of them addressing a key aspect on the adoption of energy-efficient technologies in the EU residential section. The first section is dedicated to the characterization of the stakeholder profile (Section 1). Respondents were then requested to select their building typology and project type, i.e., the "building combos", based on their last realized project (Section 2). The behavioural factors determining adoption decisions were measured in two ways: First, the respondents are asked to assess the perceived influence, interest, and level of communication with actors involved in the decision affecting the selection of the technology. Then, they were asked about the motivations behind the project (Section 3). Following, they were asked about their level of familiarity with different building technologies. Based on their answer, they were requested to identify the main drivers and barriers in relation to the technology (Section 4).

Finally, questions about contextual factors, such as the building, and sociodemographic characteristics were posed (Section 5). In this paper, only Sections 1–3 have been used for the analysis. Further details on the questionnaire can be found in Appendix A.

### 3.2. Data Analysis

### 3.2.1. The Dataset

After retrieving the data, a screening process was performed to remove responses with: (1) any of the questions used in the analysis unanswered, (2) completion time less than 8 min (as it has been the minimum tested time to complete the questionnaire), and (3) inconsistent answers (e.g., selecting all of the EERMs).

The data cleaning and processing were based on (1) the study of the response rate and (2) testing for normality. The response rate varied between countries, being the highest in Poland (31%) and the lowest in the UK (19.8%). Based on these percentages, we compared the socioeconomic composition of the respondents in the original sample. None of these percentages were deemed to raise the sampling bias. The survey responses were then filtered in order to only include demand-side actors' perspectives in the analysis, given their key role in the EE technology selection. Other stakeholders, such as architects, engineers, and construction companies can also be persuaders in this process [35]. However, they have not been included in this study. The final dataset comprises of 903 valid observations, ranging from 50 to 130 per country. Based on the previous studies on the discrete choice models, the sample size required to capture the choice should be at least 35 [36–39]. Hence, the dataset used in this study is adequate for the discrete choice modelling.

### 3.2.2. Descriptive Analysis and Model Selection

The following section presents the results of the descriptive analysis conducted to the final dataset used in estimating the discrete choice models (DCMs), along with rationale behind the selection of the modelling technique.

The objective of the survey was to portray the demand side of the market behaviour with the highest level of resolution and precision. This implied including as many EERM options as had taken place in the building project, and collecting revealed preference in lieu of stated preference. Meaning, the respondents had to answer for EERMs that had already implemented in their buildings rather than stating EERMs they assume they would implement. Table 3 also shows the eleven elements/systems and the three options (i.e., maintenance, upgrade, and new element) provided for each of these eleven elements (total of 33 measures), and their respective variable names in the model. For example, 'Maintenance_Wall' stands for the measure 'maintenance of the outer wall, and 'New_Basement' stands for the measure 'new basement/crawl space'. Each variable is binary, i.e., equals to one if the EERM is chosen and zero otherwise.

To support the analysis, it was important to select a model specification that could provide a good fit for revealed preference data and an option to choose any of the 33 different EERMs. The model specifications considered are shown in Figure 1. The multinomial logit (MNL) assumes that the user chooses just one of the EERMs, i.e., each EERM is an independent choice for the user. However, MNL model could be proved to be an unsuitable choice for the model specification as we can see that just 20% of the total observations selected only one EERM (refer Table 4). The possibility of a nested logit (NLT) model was tested, assuming the users choose one of maintenance, upgrade and new; followed by one or any combinations of 11 elements (wall, windows, roof, etc.). However, 225 (25%) observations chose all three of maintenance, upgrade, and new (irrespective of the element); similarly, 108 (12%) observations chose both maintenance and upgrade but not new; 95 (11%) observations chose upgrade and new but not maintenance; 58 (6%) observations chose maintenance and new but not upgrade. As a consequence, the MNL and NLT, which use a structure based on labelled alternatives, would have non-exclusive alternatives because more than one alternative in a choice set can be chosen. This violates the mutually exclusive assumption in discrete choice models. The levels in NLT model shown in Figure 1

could also be reversed, i.e., the users could choose one of 11 elements followed by 3 options (maintenance, upgrade, and new) as shown in Table 4. As can be depicted, in addition to multiple elements selected, there exists a handful observations selecting more than one option from maintenance, upgrade, and new for each element. For example, nearly 21 (4%) observations selected both for maintenance and the upgrade of wall. This is evident from the number of observations in dataset (903) being much smaller (nearly 1/4th) compared to the total choices in this table (3597), since an observation could choose multiple elements and choices. This shows that the choices are far away from the single choice per observation. For example, an observation could select new only in the wall and windows. This applies to two and all three choices as well. For example, an observation could select new and upgrade but not maintenance in both the wall and windows.

**Table 3.** EERMs and respective variable name in the model.

| # | Building Element or System | EEM | | |
|---|---|---|---|---|
| | | Maintenance (Including Repair) | Upgrade of Existing Elements or Systems (Incl. Insulation and Control) | New Element or Systems |
| 1 | Wall (outer) | Maintenance_Wall | Upgrade_Wall | New_Wall |
| 2 | Windows | Maintenance_Windows | Upgrade_Windows | New_Windows |
| 3 | Roof (pitched/flat) or attic | Maintenance_Roof | Upgrade_Roof | New_Roof |
| 4 | Basement/crawl space | Maintenance_Basement | Upgrade_Basement | New_Basement |
| 5 | Ventilation system | Maintenance_Ventilation | Upgrade_Ventilation | New_Ventilation |
| 6 | Energy generation (PV or solar collector) | Maintenance_EnergyGeneration | Upgrade_EnergyGeneration | New_EnergyGeneration |
| 7 | Energy storage | Maintenance_EnergyStorage | Upgrade_EnergyStorage | New_EnergyStorage |
| 8 | Appliances | Maintenance_Appliances | Upgrade_Appliances | New_Appliances |
| 9 | Heating system | Maintenance_HeatingSystem | Upgrade_HeatingSystem | New_HeatingSystem |
| 10 | Cooling system | Maintenance_CoolingSystem | Upgrade_CoolingSystem | New_CoolingSystem |
| 11 | Combined heating & cooling system | Maintenance_CombHeatingCooling | Upgrade_CombHeatingCooling | New_CombHeatingCooling |

**Figure 1.** Model selection.

Hence, the NLT model is not a valid model form as 54% of sample choose more than one choice of maintenance, upgrade, and new. After investigating the dataset in detail, the conditional logit (CL) model was found to better capture the user behaviour. CL uses unlabelled alternatives, i.e., each alternative is defined by its attributes. Additionally, unlike MNL or NLT models, CL allows to estimate the utility as a function of attributes (33 EEMs in this case) without requiring the user to have a particular choice, i.e., users can choose any combination of 33 EEMs, which is observed in the dataset. Similar to MNL, the CL model also assumes independence of irrelevant alternatives (IIA), while the choice-set (all possible alternatives) is fixed in the MNL model, whereas the attributes are fixed in the CL model. CL is flexible with any choice-set provided the alternatives in choice-set are a function of attributes, i.e., any combinations of 33 EEMs in this paper. Once the choice-set is

defined, the use of CL is exactly the same as MNL model. Further details on the modelling framework (equations) are provided in Section 3.3.

**Table 4.** Distribution of combination of choices per each element (out of 11) chosen.

| No. | Element or System | Only 1 Choice | | | Only 2 Choices | | | All 3 Choices | Total |
|---|---|---|---|---|---|---|---|---|---|
| | | Maintenance | Upgrade | New | Maintenance & Upgrade | Upgrade & New | Maintenance & New | | |
| 1 | Wall (outer) | 202 | 173 | 78 | 21 | 12 | 6 | 11 | 503 |
| 2 | Windows | 157 | 188 | 174 | 24 | 21 | 12 | 12 | 588 |
| 3 | Roof (pitched/flat) or attic | 146 | 163 | 114 | 20 | 9 | 10 | 5 | 467 |
| 4 | Basement/crawl space | 96 | 118 | 58 | 6 | 8 | 2 | 10 | 298 |
| 5 | Ventilation system | 106 | 102 | 103 | 10 | 7 | 2 | 6 | 336 |
| 6 | Energy generation | 124 | 152 | 185 | 17 | 13 | 18 | 11 | 520 |
| 7 | Energy storage | 66 | 68 | 117 | 8 | 11 | 15 | 6 | 291 |
| 8 | Appliances | 49 | 64 | 68 | 4 | 4 | 2 | 1 | 192 |
| 9 | Heating system | 31 | 41 | 87 | 8 | 3 | 3 | 0 | 173 |
| 10 | Cooling system | 25 | 41 | 44 | 0 | 2 | 2 | 0 | 114 |
| 11 | Combined heating & cooling | 22 | 50 | 39 | 1 | 1 | 1 | 1 | 115 |
| | Total | 1024 | 1160 | 1067 | 119 | 91 | 73 | 63 | 3597 |

Since the respondent was allowed to select any combination of these EERMs, it is important to see the number of EERMs out of these 33 that were chosen by various buildings across countries. Table 5 shows the distribution of number of EERMs chosen by various buildings in the sampled countries. For example, 18 observations in Belgium chose just one EERM out of 33 at a time. The number of EERMs chosen at a time varied from 1 to 25. In all countries, more than 95% of the buildings chose less than 12 EERMs at a time (in Spain it is 100%). Just less than 1% of the sample size chose each of 12–25 EERMs. This shows that the number of possible combinations of EERMs could be much less than 233, with 95% being 33 chose $i$, where $i$ was between 1 and 12 $\left( = \sum_{i=1}^{12} 33C_i \right)$. Where, $C$ is defined by N that chose r.

**Table 5.** Distribution of number of EERMs (out of 33) chosen.

| Measures Selected | Belgium | | France | | Germany | | Italy | | Netherlands | | Poland | | Spain | | UK | | Total | |
|---|---|---|---|---|---|---|---|---|---|---|---|---|---|---|---|---|---|---|
| | No | % | No | % | No | % | No | % | No | % | No | % | No | % | No | % | No | % |
| 1 | 18 | 17% | 29 | 22% | 25 | 20% | 36 | 32% | 26 | 25% | 20 | 12% | 4 | 7% | 21 | 23% | 179 | 20% |
| 2 | 10 | 10% | 27 | 20% | 18 | 14% | 36 | 32% | 19 | 18% | 29 | 17% | 5 | 9% | 16 | 17% | 160 | 18% |
| 3 | 14 | 13% | 21 | 16% | 17 | 13% | 17 | 15% | 21 | 20% | 31 | 18% | 9 | 17% | 19 | 20% | 149 | 17% |
| 4 | 6 | 6% | 12 | 9% | 19 | 15% | 10 | 9% | 7 | 7% | 13 | 8% | 10 | 19% | 12 | 13% | 89 | 10% |
| 5 | 13 | 12% | 10 | 7% | 6 | 5% | 3 | 3% | 4 | 4% | 12 | 7% | 3 | 6% | 4 | 4% | 55 | 6% |
| 6 | 13 | 12% | 5 | 4% | 8 | 6% | 5 | 4% | 1 | 1% | 13 | 8% | 6 | 11% | 2 | 2% | 53 | 6% |
| 7 | 4 | 4% | 1 | 1% | 2 | 2% | | 0% | 6 | 6% | 10 | 6% | 1 | 2% | 1 | 1% | 25 | 3% |
| 8 | 9 | 9% | 8 | 6% | 7 | 6% | 3 | 3% | 6 | 6% | 17 | 10% | 7 | 13% | 4 | 4% | 61 | 7% |

**Table 5.** *Cont.*

| Measures Selected | Belgium | | France | | Germany | | Italy | | Netherlands | | Poland | | Spain | | UK | | Total | |
|---|---|---|---|---|---|---|---|---|---|---|---|---|---|---|---|---|---|---|
| | No | % | No | % | No | % | No | % | No | % | No | % | No | % | No | % | No | % |
| 9 | 2 | 2% | 6 | 4% | 6 | 5% | | 0% | 2 | 2% | 8 | 5% | 4 | 7% | 4 | 4% | 32 | 4% |
| 10 | 2 | 2% | 4 | 3% | 2 | 2% | 1 | 1% | 2 | 2% | 4 | 2% | 5 | 9% | 3 | 3% | 23 | 3% |
| 11 | 12 | 11% | 9 | 7% | 12 | 9% | 1 | 1% | 9 | 8% | 4 | 2% | | 0% | 3 | 3% | 50 | 6% |
| 12 | 1 | 1% | | 0% | 1 | 1% | | 0% | 2 | 2% | 5 | 3% | | 0% | 2 | 2% | 11 | 1% |
| 13 | 1 | 1% | | 0% | 1 | 1% | | 0% | 1 | 1% | 1 | 1% | | 0% | | 0% | 4 | 0% |
| 14 | | 0% | 1 | 1% | 1 | 1% | 1 | 1% | | 0% | 3 | 2% | | 0% | 1 | 1% | 7 | 1% |
| 15 | | 0% | | 0% | 1 | 1% | | 0% | | 0% | | 0% | | 0% | 1 | 1% | 2 | 0% |
| 19 | | 0% | | 0% | | 0% | | 0% | | 0% | 1 | 1% | | 0% | | 0% | 1 | 0% |
| 24 | | 0% | | 0% | 1 | 1% | | 0% | | 0% | | 0% | | 0% | | 0% | 1 | 0% |
| 25 | | 0% | 1 | 1% | | 0% | | 0% | | 0% | | 0% | | 0% | | 0% | 1 | 0% |
| Total | 105 | | 134 | | 127 | | 113 | | 106 | | 171 | | 54 | | 93 | | 903 | |

The percentage of observations choosing each EERM, where an observation/building could choose multiple EERMs (hence, the rows in Table 6 did not add up to 100%), is presented in Table 6. In the overall sample, the most frequently chosen EERMs (>20%) were: maintenance of wall, windows, roof (Maintainance_Wall, Maintainance_Windows, and Maintainance_Roof), upgrade of wall, windows, roof, energy generation (Upgrade_Wall, Upgrade_windows, Upgrade_roof, and Upgrade_EnergyGeneration) and new windows, and energy generation (New_Windows and New_EnergyGeneration). The least frequent EERMs (<10%) were: maintenance and upgrade of appliances, heating, cooling, and combined heating and cooling system (Maintenance_Appliances, Maintenance_HeatingSystem, Maintenance_Cooling, Maintenance_CombHeatingCooling, Upgrade_HeatingSystem, Upgrade_CoolingSystem, Upgrade_CoolingSystem, and Upgrade_CombHeatingCooling), and new installation of basement, appliances, cooling, and combined heating and cooling systems (New_Basement, New_Appliances, New_CoolingSystem, and New_ CombHeatingCooling). Results show a large variation in the frequency of EERMs chosen among the countries. The EERMs that tops the list in each country were: Maintenance of wall (Maintenance_Wall) in France and Poland; new windows (New_Windows) in Germany; upgrade of windows (Upgrade_Windows) in Italy and UK; and new energy generation (New_EnergyGeneration) in Belgium and Spain. From these descriptive statistics, it is evident that the DCMs should show different behaviour in various countries.

**Table 6.** Distribution of sample size and percentage of observations choosing each EERM (of 33).

| EEM | Belguim | | France | | Germany | | Italy | | Netherlands | | Poland | | Spain | | UK | | Total | |
|---|---|---|---|---|---|---|---|---|---|---|---|---|---|---|---|---|---|---|
| | No | % | No | % | No | % | No | % | No | % | No | % | No | % | No | % | No | % |
| Maintenance_Wall | 24 | 23% | 38 | 28% | 38 | 30% | 20 | 18% | 21 | 20% | 56 | 33% | 18 | 33% | 25 | 27% | 240 | 27% |
| Maintenance_Windows | 22 | 21% | 36 | 27% | 28 | 22% | 21 | 19% | 20 | 19% | 46 | 27% | 6 | 11% | 26 | 28% | 205 | 23% |
| Maintenance_Roof | 14 | 13% | 26 | 19% | 22 | 17% | 24 | 21% | 16 | 15% | 46 | 27% | 11 | 20% | 22 | 24% | 181 | 20% |
| Maintenance_Basement | 17 | 16% | 17 | 13% | 15 | 12% | 4 | 4% | 8 | 8% | 29 | 17% | 17 | 31% | 7 | 8% | 114 | 13% |
| Maintenance_Ventilation | 17 | 16% | 23 | 17% | 16 | 13% | 1 | 1% | 14 | 13% | 38 | 22% | 2 | 4% | 13 | 14% | 124 | 14% |
| Maintenance_EnergyGeneration | 20 | 19% | 28 | 21% | 31 | 24% | 11 | 10% | 16 | 15% | 43 | 25% | 4 | 7% | 17 | 18% | 170 | 19% |
| Maintenance_EnergyStorage | 9 | 9% | 17 | 13% | 16 | 13% | 7 | 6% | 7 | 7% | 27 | 16% | 2 | 4% | 10 | 11% | 95 | 11% |
| Maintenance_Appliances | 7 | 7% | 10 | 7% | 9 | 7% | 1 | 1% | 7 | 7% | 15 | 9% | 1 | 2% | 6 | 6% | 56 | 6% |
| Maintenance_HeatingSystem | 3 | 3% | 11 | 8% | 4 | 3% | 3 | 3% | 8 | 8% | 7 | 4% | 1 | 2% | 5 | 5% | 42 | 5% |

**Table 6.** *Cont.*

| EEM | Belguim | | France | | Germany | | Italy | | Netherlands | | Poland | | Spain | | UK | | Total | |
|---|---|---|---|---|---|---|---|---|---|---|---|---|---|---|---|---|---|---|
| | No | % | No | % | No | % | No | % | No | % | No | % | No | % | No | % | No | % |
| Maintenance_CoolingSystem | 1 | 1% | 4 | 3% | 3 | 2% | 3 | 3% | 7 | 7% | 4 | 2% | 1 | 2% | 4 | 4% | 27 | 3% |
| Maintenance_CombHeatingCooling | 1 | 1% | 7 | 5% | 4 | 3% | 2 | 2% | 4 | 4% | 3 | 2% | 1 | 2% | 3 | 3% | 25 | 3% |
| Upgrade_Wall | 29 | 28% | 31 | 23% | 28 | 22% | 24 | 21% | 24 | 23% | 56 | 33% | 12 | 22% | 13 | 14% | 217 | 24% |
| Upgrade_Windows | 32 | 30% | 38 | 28% | 29 | 23% | 27 | 24% | 33 | 31% | 41 | 24% | 18 | 33% | 27 | 29% | 245 | 27% |
| Upgrade_Roof | 23 | 22% | 27 | 20% | 25 | 20% | 24 | 21% | 27 | 25% | 40 | 23% | 12 | 22% | 19 | 20% | 197 | 22% |
| Upgrade_Basement | 17 | 16% | 18 | 13% | 27 | 21% | 9 | 8% | 19 | 18% | 36 | 21% | 5 | 9% | 11 | 12% | 142 | 16% |
| Upgrade_Ventilation | 21 | 20% | 21 | 16% | 14 | 11% | 3 | 3% | 17 | 16% | 30 | 18% | 6 | 11% | 13 | 14% | 125 | 14% |
| Upgrade_EnergyGeneration | 35 | 33% | 26 | 19% | 24 | 19% | 11 | 10% | 26 | 25% | 35 | 20% | 10 | 19% | 26 | 28% | 193 | 21% |
| Upgrade_EnergyStorage | 9 | 9% | 12 | 9% | 13 | 10% | 17 | 15% | 19 | 18% | 14 | 8% | 0 | 0% | 9 | 10% | 93 | 10% |
| Upgrade_Appliances | 13 | 12% | 14 | 10% | 14 | 11% | 2 | 2% | 12 | 11% | 10 | 6% | 0 | 0% | 8 | 9% | 73 | 8% |
| Upgrade_HeatingSystem | 9 | 9% | 9 | 7% | 10 | 8% | 1 | 1% | 12 | 11% | 6 | 4% | 1 | 2% | 4 | 4% | 52 | 6% |
| Upgrade_CoolingSystem | 8 | 8% | 7 | 5% | 8 | 6% | 6 | 5% | 7 | 7% | 4 | 2% | 3 | 6% | 0 | 0% | 43 | 5% |
| Upgrade_CombHeatingCooling | 11 | 10% | 9 | 7% | 9 | 7% | 8 | 7% | 6 | 6% | 5 | 3% | 0 | 0% | 5 | 5% | 53 | 6% |
| New_Wall | 22 | 21% | 10 | 7% | 18 | 14% | 3 | 3% | 6 | 6% | 22 | 13% | 14 | 26% | 12 | 13% | 107 | 12% |
| New_Windows | 25 | 24% | 32 | 24% | 43 | 34% | 15 | 13% | 16 | 15% | 45 | 26% | 24 | 44% | 19 | 20% | 219 | 24% |
| New_Roof | 24 | 23% | 10 | 7% | 27 | 21% | 3 | 3% | 17 | 16% | 29 | 17% | 12 | 22% | 16 | 17% | 138 | 15% |
| New_Basement | 12 | 11% | 8 | 6% | 10 | 8% | 3 | 3% | 8 | 8% | 18 | 11% | 10 | 19% | 9 | 10% | 78 | 9% |
| New_Ventilation | 21 | 20% | 19 | 14% | 15 | 12% | 2 | 2% | 11 | 10% | 29 | 17% | 14 | 26% | 7 | 8% | 118 | 13% |
| New_EnergyGeneration | 26 | 25% | 26 | 19% | 42 | 33% | 13 | 12% | 23 | 22% | 54 | 32% | 25 | 46% | 18 | 19% | 227 | 25% |
| New_EnergyStorage | 16 | 15% | 13 | 10% | 35 | 28% | 23 | 20% | 12 | 11% | 31 | 18% | 11 | 20% | 8 | 9% | 149 | 17% |
| New_Appliances | 12 | 11% | 7 | 5% | 15 | 12% | 4 | 4% | 6 | 6% | 15 | 9% | 8 | 15% | 8 | 9% | 75 | 8% |
| New_HeatingSystem | 22 | 21% | 11 | 8% | 13 | 10% | 1 | 1% | 10 | 9% | 14 | 8% | 15 | 28% | 7 | 8% | 93 | 10% |
| New_CoolingSystem | 11 | 10% | 6 | 4% | 4 | 3% | 3 | 3% | 2 | 2% | 4 | 2% | 17 | 31% | 1 | 1% | 48 | 5% |
| New_CombHeatingCooling | 9 | 9% | 6 | 4% | 11 | 9% | 4 | 4% | 4 | 4% | 5 | 3% | 0 | 0% | 3 | 3% | 42 | 5% |

### 3.3. Modelling Approach

In formulating a statistical model for ordered discrete outcomes, it is common to start with a linear function of covariates that influence specific discrete results [40]. Due to the wide number of answer options in the survey question, this is undertaken with the help of a conditional logit (CL) model based on the random utility theory (RUT). The RUT is based on the hypothesis that the users (choice-makers) are rational individuals who try to maximize the perceived utility from the choice made. Hence, the probability that an alternative $k$ is selected is equal to the probability that the utility derived by the user in choosing $k$ is higher than utilities of choosing any other alternatives apart from $k$ (see (7)). The utilities could be measured as a linear function of attributes that are either relevant to the choice or the user. In this research, the utilities of making different choices of EERMs are estimated as a function of country, building combo, and motivation behind the project as explained in (2) to (6). The objective of the CL is to estimate a function that determines outcome probabilities. In this case, the probability of a certain choice of EERMs to take place given a specific set of choices, country and motivation behind the building combo. To evaluate how the probabilities might vary across countries, these have been identified as the main effect in the model. In the model, the building envelope elements have been assessed separately from the heating/cooling elements and appliances based on the different life-cycles each of these groups have, and the distinct potential motivations behind retrofitting them [41].

### 3.3.1. Model Formulations: Main Effects

These formulations capture the overall effect of EERMs, by either country $c$, or building combo $b$, or motivation $m$ in selecting an alternative $k$ from a given choice-set (one or combination of various EERMs, $i$) using the CL model. The CL model estimates utility

as a function of attributes (EERMs) and does not consider any particular alternative or a choice-set (set of alternatives). Hence, the CL could estimate the utility for any given choice-set. For example, a choice set could be a set of 33 EERMs, where only one EERM could be selected at a time, i.e., 33 alternatives with each EERM being an alternative. The CL model could estimate the probability of selecting each EERM (an alternative). Since the CL needs to capture the behaviour with respect to any given choice-set, which is a combination of EERMs, there is no utility without an EERM being an independent variable (binary). Hence, the binary variable for an EERM ($X_i$) is present in every utility function of the CL model as shown below. Simply, without an EERM there is no alternative to select from, without an alternative there is no utility of selecting it, therefore the presence of Xi in each utility function is explained below. Therefore, based on the findings from the data analysis presented in Section 3.2, it is assumed that the user could choose any combination of 33 EERMs for a choice-set and each alternative in the choice-set is independent. The choice-set ($K$) with $n$ alternatives is defined in (1) below. Where, each alternative ($k_i$) is a subset of the 33 EERMs presented in Table 3.

$$K = \{k_1, k_2, \ldots k_n\}, \ \ each \ k_i \complement \{X_1, X_2, \ldots X_{33}\}, \ k_i \neq k_j \forall i, j; n \ is \ any \ positive \ integer \quad (1)$$

The utility of selecting an alternative $k$ from a given choice-set (defined in (1)), is defined by ($U_k$), by an observation $n$ (subscript ignored in the equation for simplicity) with various main effects are shown in (2)–(5). These main effect models estimate the parameters of $U_k$ ($\alpha_i$, $\alpha_{ci}$, $\alpha_{bi,}$ and $\alpha_{mi}$), based on the below hypothesis.

Null hypothesis, H0: The influence of EERMs on the selection an alternative $k$ from a given choice-set (defined in (1)) significantly vary either by country c, or building combo b, or motivation m

Alternative hypothesis, H1:H0 is not true.

EERM model: Estimates the CL model for the entire dataset, assuming EERMs are the only independent variable.

$$U_k = \sum_{i \in EEM} (\alpha_i X_i) + \varepsilon_k \forall k \in K \quad (2)$$

Country main effect model: Estimates the CL model for the entire dataset, assuming EERMs and country are the only independent variables.

$$U_k = \sum_{i \in EEM} \delta_c \alpha_{ci} X_i + \varepsilon_k \forall k \in K, c \in Country \quad (3)$$

Building combo main effect model: Estimates the CL model for the entire dataset, assuming EERMs and building combo are the only independent variables.

$$U_k = \sum_{i \in EEM} \delta_b \alpha_{bi} X_i + \varepsilon_k \forall k \in K, b \in Bucket \quad (4)$$

Motivation main effect model: Estimates the CL model for the entire dataset, assuming EERMs and motivation are the only independent variables.

$$U_k = \sum_{i \in EEM} \delta_m \alpha_{mi} X_i + \varepsilon_k \forall k \in K, m \in Motivation \quad (5)$$

where,

$k$ = the choice from a given choice-set defined in (1).
$X_i$ = is the binary variable for 33 EERMs, mentioned in Table 3.
$\delta_c$ = is the binary variable for the 8 countries, mentioned in Table 7.
$\delta_b$ = is the binary variable for the 4 building combo, mentioned in Table 7.
$\delta_m$ = is the binary variable for the 5 motivations, mentioned in Table 7.
$\alpha_i$, $\alpha_{ci}$, $\alpha_{bi,}$ and $\alpha_{mi}$ are the model parameters.

**Table 7.** Definition of binary variables (and respective variable names).

| Country ($\delta_c$) | Building Combo ($\delta_b$) | Motivation ($\delta_m$) |
|---|---|---|
| Italy (IT) | Retrofit of Single-dwelling building (R_SDB) | Environmental (Env) |
| Spain (ES) | Retrofit of Multidwelling building (R_MDB) | Technical (Tech) |
| Poland (PL) | Deep retrofit of Single-dwelling building (D_SDB) | Economic (Eco) |
| Germany (DE) | Deep retrofit of Multidwelling building (D_MDB) | Social (Soc) |
| Netherlands (NL) | | Legal (Leg) |
| United Kingdom (UK) | | |
| France (FR) | | |
| Belgium (BE) | | |

### 3.3.2. Model Formulations: Combined All Main Effects

The combined utility of selecting an alternative $k$ from a given choice-set (defined in (1)), is defined by ($U_k$), by an observation $n$ (subscript ignored in the equation for simplicity), in country $c$, and in building combo $b$, and with motivation $m$, is described in (6):

$$U_k = \sum_{i \in EEM} \left( \sum_{c \in Country} \delta_c \beta_{ci} X_i + \sum_{b \in Bucket} \delta_b \beta_{bi} X_i + \sum_{m \in Motivation} \delta_m \beta_{mi} X_i \right) + \varepsilon_k \forall k \in K \tag{6}$$

where, $\beta_{ci}$, $\beta_{bi}$, and $\beta_{mi}$ are the model parameters. It could be noted that there is a total of 561 (=33 × (8 + 4 + 5)) parameters to be estimated by the logit model.

The model in (6) is different from (2) to (5) as the former model could capture the effect of country, building combo, and motivation simultaneously while the latter models could analyse just one of country, building combo, and motivation at a time. For example, (3) could be used to compare the probabilities of selecting an EERM across different countries but cannot be used to study the same across different building combos. Similarly, (4) helps in studying the choice of EERMs exclusively across building combos. Whereas, (6) could quantify the choice of EERMs at the disaggregated level such as what is the probability of an EERM at a given country, building combo, and motivation of the project, e.g., in Spain (ES) belonging to the retrofit of single-dwelling building (R_SDB), with an environmental (Env) motivation? The addition of the "$\varepsilon_k$" vector of errors (also called disturbance term) is supported on a number of grounds such as the possibility that some potential variables that influence the choice could have been omitted from the equation [40].

The definition of the binary variables: building combo, motivations, and countries are described in Table 7. The EERMs have been previously described in Table 3. The country and building combo variables are collectively exhaustive, i.e., a given observation belongs to one of the eight countries and four building combo each. Whereas motivations are separate binary variables. For example, if an observation chooses environmental then the variable Env = 1, otherwise Env = 0. This is similar, for the other four motivation variables. As mentioned in Section 3.2 an observation is allowed to select multiple motivations.

### 3.3.3. Model Formulations: Probability and Elasticities

The probability that an observation $n$ selecting an alternative $k$ in a choice-set in (1) is given by the Equation (2), where $U_k$ is defined in Equations (1)–(6).

$$P_k = \frac{exp(U_k)}{\sum_{k \in K} exp(U_k)} \tag{7}$$

The elasticities with respect to an attribute ($X_i$) vary with the choice-set. Hence, the elasticities cannot be estimated from the dataset along with the CL model estimation, as the dataset does not have a specific choice-set. In this paper, the elasticity with respect to an EERM ($X_i$) depends on the number of alternatives in the choice-set ($K$), which contain the EERM ($X_i$). Equations (8) and (9) show the elasticities for (6), where $k'$ in $k \in k\prime$ is the set of all alternatives, which contain $X_i$. (Note: since $X_i$ is binary, could only take the values either 0 or 1). In this context, the elasticities in (8) and (9) estimate the relative change in the probability of selecting an EERM in a country, building combo, and motivation of the

project if an EERM is included or excluded in the choicest. For example, what would be percentage increase in the probability of selecting the maintenance of wall if installing new wall is not a possibility anymore?

$$\frac{\partial(P_k)/P_k}{\partial(X_i)/X_i} = \left( \sum_{c \in Country} \delta_c \beta_{ci} + \sum_{b \in Bucket} \delta_b \beta_{bi} + \sum_{m \in Motivation} \delta_m \beta_{mi} \right) X_i \left( 1 - \sum_{k \in k\prime} \delta_m P_k \right) \; if \; X_i \in k \tag{8}$$

$$\frac{\partial(P_k)/P_k}{\partial(X_i)/X_i} = - \left( \sum_{c \in Country} \delta_c \beta_{ci} + \sum_{b \in Bucket} \delta_b \beta_{bi} + \sum_{m \in Motivation} \delta_m \beta_{mi} \right) X_i \left( \sum_{k \in k\prime} \delta_m P_k \right) \; if \; X_i \notin k \tag{9}$$

## 4. Model Results

### 4.1. Logit Model

The CL model results are presented in Tables 8 and 9. The main effect models explained in Equations (2)–(5) are shown in Table 8. In parallel, Table 9 shows the combined main effects model explained in Equation (6), while entire Table 9 is a single model.

Each column in the Table 8 represents an individual model. The column under the header 'Total' presents the EERM model parameters ($\alpha_i$) in Equation (2), columns under country ($\delta_c$) present $\alpha_{ci}$ in (3), columns under building combo ($\delta_b$) present $\alpha_{bi}$ in (4), and columns under motivation ($\delta_m$) present $\alpha_{mi}$ in (5). All models contained parameters that were significant at the 5% level and display a good fit with pseudo $R^2$ ranging between 0.85 and 0.96, probability greater than chi2 close to zero. The blanks indicate the parameters, which were not significantly different from zero. All significant parameters were negative, which shows that the utility of these EERMs was lower than the other EERMs with zero values. For example, in the case of Spain, the utility of selecting new ventilation was lower than selecting any other new elements (wall, windows, etc.). Additionally, the parameter values cannot be translated to the choice of EERMs, as the probabilities depend on the choice-set. For instance, if the choice-set had just two alternatives (new wall and new window) in Spain, the probability of selecting them was 0.5 each, since both had zero parameters. In the 'Total' model, except for 9 EERMs, 24 EERMs were statistically significant. The significance of various EERMs changed among the countries and building combo. For example, in countries Spain and Belgium, the maintenance windows negatively influence the utility of a choice *k* and not significant in other six countries. However, the motivations show some similar trends among them. The economic (*Eco*) and social (*Soc*), environmental (*Env*), and legal (*Leg*) motivations had almost same significant EERMs and parameter estimates. This shows that for any given choice-set the probability of selecting an alternative did not differ considerably between '*Eco*' and '*Soc*' and '*Env*' and '*Leg*' motivations.

**Table 8.** Logit model results: main effects (Equations (2)–(5)).

| Group | $X_i$ | Total | Country ($\delta_c$) | | | | | | | | Building Combo ($\delta_b$) | | | | Motivation ($\delta_m$) | | | | |
|---|---|---|---|---|---|---|---|---|---|---|---|---|---|---|---|---|---|---|---|
| | | | ES | IT | PL | DE | NL | UK | FR | BE | R_SDB | R_MDB | D_SDB | D_MDB | Env | Eco | Soc | Tech | Leg |
| Maintenance_ | Wall | −0.97 | −1.97 | | | | −3.08 | | | −4.52 | −3.21 | | | −1.68 | | | | | |
| | Windows | | −3.39 | | | | | | | −5.33 | | | | −2.52 | | | | | |
| | Roof | | | | | | −3.24 | −2.97 | | −4.24 | | | | | | | | | |
| | Basement | −1.13 | | | | | | | | | | | −1.78 | | | | −0.93 | −0.93 | |
| | Ventilation | −0.74 | | | | −1.54 | | | | −4.00 | −5.63 | | | −1.50 | −0.88 | | | −0.96 | −0.95 |
| | EnergyGeneration | | | | | | −4.64 | | | | | −1.53 | −2.59 | | −0.83 | | | −0.82 | −1.24 |
| | EnergyStorage | −1.14 | | | | −1.49 | | | | −3.37 | | | −2.87 | | −1.49 | −1.32 | −1.32 | −1.20 | −1.51 |
| | Appliances | −1.66 | | | −2.58 | | −3.63 | −3.29 | −2.16 | −6.46 | | | | | −1.77 | −1.80 | −1.80 | −1.73 | −1.84 |
| | Heating | −2.06 | | −5.06 | −3.28 | −3.62 | −2.97 | | | | | −4.61 | −6.10 | −5.28 | −1.64 | −1.56 | −1.56 | −1.60 | −1.68 |
| | Cooling | −2.63 | | −7.14 | −5.03 | −2.82 | | −3.23 | −2.32 | | | −4.31 | −4.03 | −5.34 | −2.06 | −2.20 | −2.20 | −2.14 | −1.89 |
| | CombHeatingCooling | −2.79 | | −8.02 | −3.52 | −3.72 | −2.58 | −6.01 | −4.26 | | | −8.18 | | −3.59 | −2.96 | −2.84 | −2.84 | −2.60 | −2.47 |
| Upgrade_ | Wall | −0.90 | −1.97 | | | | | | | | | | −1.61 | | | −0.81 | −0.81 | | −1.02 |
| | Windows | −0.75 | | | | −1.60 | | | | −2.46 | | −2.37 | | −1.93 | | −0.76 | −0.76 | | −1.06 |
| | Roof | −0.93 | −1.74 | | | | | | −2.91 | | | | | −1.87 | −1.03 | −0.94 | −0.94 | −0.78 | −0.91 |
| | Basement | | −5.37 | −4.03 | | | −3.80 | | | −3.58 | | | | | | | | | |
| | Ventilation | −0.77 | −5.01 | | | | −2.91 | | | −3.50 | | | | | −0.99 | | | −1.16 | |
| | EnergyGeneration | | −1.90 | | −2.19 | | −4.28 | | | | | | | | −2.88 | | | | |
| | EnergyStorage | −1.38 | | | −3.01 | | | | −1.77 | −5.49 | | −4.03 | −5.53 | −1.51 | −0.75 | −1.04 | −1.04 | −0.96 | −1.19 |
| | Appliances | −1.30 | | | | | | | | | | | | −4.39 | −1.17 | −1.36 | −1.36 | −1.12 | −1.06 |
| | Heating | −1.56 | | | −2.56 | −1.67 | −2.55 | | −2.10 | −3.76 | −6.27 | −3.39 | −3.64 | −3.19 | −1.57 | −1.47 | −1.47 | −1.62 | −1.45 |
| | Cooling | −2.00 | | | | −3.71 | | | −2.51 | −6.93 | −9.68 | −3.41 | −3.84 | −1.77 | −2.61 | −2.26 | −2.26 | −2.36 | −2.65 |
| | CombHeatingCooling | −1.67 | | −7.32 | | −3.23 | −5.98 | | −3.38 | | | | −8.63 | | −2.29 | −1.47 | −1.47 | −1.88 | −1.89 |
| New_ | Wall | −1.68 | −2.48 | −6.34 | | | | −3.97 | | −2.89 | −5.87 | | −2.74 | | −1.89 | −1.21 | −1.21 | −1.63 | −2.11 |
| | Windows | −1.37 | | −4.05 | | | | | | | | | | | −1.65 | −0.95 | −0.95 | −1.39 | −1.73 |
| | Roof | | −1.56 | | | −2.99 | −3.01 | | −3.09 | | | | | −2.47 | −1.61 | −1.05 | −1.05 | | |
| | Basement | −1.48 | −1.76 | | −4.62 | −3.77 | −4.55 | | −2.47 | −4.96 | −8.01 | | | −2.27 | −1.30 | −1.42 | −1.42 | −1.07 | −1.24 |
| | Ventilation | | −2.06 | | | | −4.79 | | −2.19 | | | | | | −1.07 | | | −1.06 | |
| | EnergyGeneration | | | | | | | | | | | | | | | | | | |
| | EnergyStorage | | | | −1.95 | | −4.30 | −3.67 | | −4.22 | −4.51 | | | | | | | | |
| | Appliances | −1.65 | −1.91 | −3.17 | | −2.66 | | −4.05 | | | | | −1.68 | −1.71 | −1.65 | −1.55 | −1.55 | −1.58 | −2.29 |
| | Heating | −0.82 | | | | | | | | | | | | −2.06 | | | | | |
| | Cooling | −2.16 | | | −5.81 | | −7.13 | | | | | | | −2.13 | −2.77 | −2.15 | −2.15 | −2.48 | −2.14 |
| | CombHeatingCooling | −2.17 | | −3.58 | | | −2.50 | | −3.43 | −4.28 | | | | −3.16 | −2.45 | −1.96 | −1.96 | −2.34 | −1.74 |

**Table 8.** *Cont.*

| $X_i$ | Total | Country ($\delta_c$) | | | | | | | | Building Combo ($\delta_b$) | | | | Motivation ($\delta_m$) | | | | |
|---|---|---|---|---|---|---|---|---|---|---|---|---|---|---|---|---|---|---|
| | | ES | IT | PL | DE | NL | UK | FR | BE | R_SDB | R_MDB | D_SDB | D_MDB | Env | Eco | Soc | Tech | Leg |
| Pseudo R$^2$ | 0.96 | 0.85 | 0.96 | 0.95 | 0.92 | 0.94 | 0.94 | 0.92 | 0.93 | 0.98 | 0.92 | 0.95 | 0.90 | 0.95 | 0.95 | 0.93 | 0.95 | 0.92 |
| Log likelihood | −82.87 | −18.94 | −10.26 | −20.45 | −23.65 | −15.21 | −12.39 | −24.15 | −17.97 | −14.60 | −19.76 | −18.55 | −22.87 | −63.76 | −78.10 | −74.20 | −67.31 | −50.44 |
| Prob > chi2 | 0.00 | 0.00 | 0.00 | 0.00 | 0.00 | 0.00 | 0.00 | 0.00 | 0.00 | 0.00 | 0.00 | 0.00 | 0.00 | 0.00 | 0.00 | 0.00 | 0.00 | 0.00 |
| LR chi2 | 3992.72 | 210.79 | 499.87 | 746.57 | 537.55 | 457.73 | 403.49 | 568.80 | 447.61 | 1352.36 | 434.82 | 667.48 | 405.57 | 2520.45 | 2745.05 | 2066.69 | 2550.19 | 1230.01 |
| Obs | 903 | 54 | 113 | 171 | 127 | 106 | 93 | 134 | 105 | 300 | 103 | 153 | 98 | 575 | 630 | 481 | 583 | 289 |

**Table 9.** Logit model results: combined effects (Equation (6)).

| | Xi | Country ($\delta_c$) | | | | | | | | Building Combo ($\delta_b$) | | | | Motivation ($\delta_m$) | | | | |
|---|---|---|---|---|---|---|---|---|---|---|---|---|---|---|---|---|---|---|
| | | ES | IT | PL | DE | NL | UK | FR | BE | R_SDB | R_MDB | D_SDB | D_MDB | Env | Eco | Soc | Tech | Leg |
| Maintenance_ | Wall | | | | | −5.25 | | | −5.67 | | | | | | | | | |
| | Windows | | | | | | | | −6.92 | | | −6.01 | | | | | | |
| | Roof | −5.26 | | | | −6.40 | −8.25 | | | | | | | | | | | |
| | Basement | | | | | | | | | | | | | −1.84 | | | | |
| | Ventilation | | | | | | | −4.19 | −3.43 | | | | | | | −2.05 | | |
| | EnergyGeneration | | | | | −5.28 | | | | | | −9.66 | | | | | | −3.42 |
| | Appliances | | | | | −6.19 | | −7.93 | | | | | | | | −3.18 | | |
| | Heating | | | | | | | | | | | −7.67 | | | −6.55 | | | |
| | Cooling | | | −7.49 | −8.26 | −5.88 | | | | | | | | | | | | |
| | CombHeatingCooling | | −11.79 | −6.52 | | | −12.48 | −11.77 | | | | | | | | | | |
| Upgrade_ | Windows | | | | | | | | −5.23 | | | | | | | | −2.36 | |
| | Roof | | | | | | | | | | | | | −3.07 | | | | |
| | Basement | −4.93 | −5.38 | | | | −7.33 | | −6.54 | | | | | | | | | |
| | Ventilation | −4.90 | | | | 5.08 | | | −9.09 | | | | | | | −2.38 | | |
| | EnergyGeneration | | | | | −9.12 | | | | | | | −4.67 | | | | | |
| | EnergyStorage | | | −3.44 | | | | | | | −5.01 | −6.41 | | | | −4.70 | | 3.56 |
| | Appliances | | | | | | | | | | | −8.23 | | | | −2.09 | | |
| | Heating | | | | | −5.34 | −3.26 | −6.32 | | | | | | | | | −4.17 | |
| | Cooling | | | | | | −4.68 | −11.40 | | −7.76 | | −9.75 | −3.29 | | | | | |
| | CombHeatingCooling | | | | | −7.60 | −10.07 | | | | | | | | | | | |

**Table 9.** *Cont.*

| $Xi$ | | Country ($\delta_c$) | | | | | | | | Building Combo ($\delta_b$) | | | | Motivation ($\delta_m$) | | | | |
|---|---|---|---|---|---|---|---|---|---|---|---|---|---|---|---|---|---|---|
| | | ES | IT | PL | DE | NL | UK | FR | BE | R_SDB | R_MDB | D_SDB | D_MDB | Env | Eco | Soc | Tech | Leg |
| | Wall | | | | | | | | | | | | | −3.06 | −3.08 | | | |
| | Windows | | −3.33 | | | | | | | | | | | −3.45 | | | | |
| | Roof | −4.93 | | | | −6.01 | | −5.16 | | | | | | −3.29 | | | | |
| New_ | Basement | | | −3.43 | −3.89 | −11.82 | | | −11.24 | | | | | | | | | |
| | Ventilation | | | | | −4.82 | | | | | | | | | | | | |
| | EnergyStorage | | | | | | −5.40 | | −8.68 | −3.68 | | | | | | | | |
| | Appliances | | | | | | | | | | | | | | | | −4.95 | |
| | Heating | | | | | | | | | | | | | −5.26 | | | | |
| | Cooling | | | −7.04 | | | −14.60 | | | | | | | | | | | |
| | CombHeatingCooling | | | | | | | | | | | | | | | | | |

Obs: 903, LR chi2(77): 4007.82, Prob > chi2: 0, Pseudo R$^2$: 0.96, Log likelihood −75.32.

In Table 9, based on the model formulation, each column in the table shows the parameters of country ($\beta_{ci}$), building combo ($\beta_{bi}$), and motivation ($\beta_{mi}$), shown in Equation (6). As depicted in Table 9, just 77 out of 561 coefficients ($\beta_{ci}$, $\beta_{bi}$, and $\beta_{mi}$) were found to be significantly different from 0 at the 5% level. The blank cells in Table 9 indicate the parameters that were not significantly different from zero. In the table, the higher the value is, the higher the utility. Thus, if the value is negative, it has a low utility. For example, EERMs with the lowest utility were: the new cooling system ($\beta_{ci}$ = −14.60) and maintenance of the combined cooling and heating system ($\beta_{ci}$ = −12.48), both in the UK. Additionally, the EERM with the highest utility were: the upgrade of the ventilation system in the Netherlands ($\beta_{ci}$ = 5.08) and the upgrade of the energy storage if the motivation is legal ($\beta_{mi}$ = 3.56). Comparing Tables 8 and 9, the combined effect of country, building combo, and motivation played a significant role in the selection of an alternative in a given choice-set ($K$). For example, the EERM new window was significant when Belgium was considered alone as shown in Table 8. Whereas, Table 9 shows that the new window in Belgium was significant only if the motivation was '*Env*'. In almost all five motivations many EERMs became insignificant when the combined with country and building combo was considered. This shows that the combined model in Equation (6), i.e., Table 9, which considers all the main effects, was better in depicting the user behaviour than the models in Table 8. Hence, the next section uses the model in Table 9 to estimate the probabilities of selecting various alternatives of EERMs for a numerical example.

*4.2. Numerical Analysis*

This section aims to evaluate the influence of country, building combo, and motivation on selecting a choice of EERMs. In order to perform this, a definitive choice-set is required as the dependent variable as described in the model in 0, which is the probability of selecting an alternative for a given choice-set (combination of EERMs). The choice-set used in this section comprises a comprehensive list of 39 alternatives (see Table 10), broken down into 63 possible unique combination of EERMs, i.e., the choice set $K$ for the CL model comprises of 63 alternatives, which was developed based on the most cost-effective combinations to reach nZEB and/or cost-optimal for different building types and EU climatic zones according to Zhangheri et al. [42]. It is important to note that the single measures alone are not perceived as measures to reach nZEB standards, only in combination with others. Yet they were included in the choice set as based on previous studies indicated that these might be often undertaken measures in residential buildings in the EU [13]. These 63 combinations of EERMs represent the choice-set ($K$) explained in Equation (1). The list was then validated through discussions with market experts for each country (energy specialists with more than ten years of experience in the building sector in the given country). It is therefore deemed as a comprehensive and consistent list of EERMs entailing most of the possible actions that can be performed to a single building related to its energy efficiency.

Using the results of the modelling (Table 9), the probabilities were estimated for each alternative in the choice-set, for all possible combinations of 4 building combo, 5 motivations, and 8 countries mentioned in Table 3, i.e., a total of 160 (4 × 5 × 8) cases. The utilities along with respective probabilities were estimated using Table 9 for the choice-set (63 possible unique combinations of EERMs), then aggregated for the 39 alternatives. The aggregation simply requires summing up the probabilities for the combinations in each of the 39 alternatives. Once the probabilities are estimated for 39 alternatives for 160 cases, for each case, the 39 alternatives were ranked in the ascending order of their probabilities. The rank 39 is given for an alternative with the highest probability and rank 1 for the least. The alternatives with equal probabilities were given the same rank. Then, for each alternative in the 39 choices, the average value of the ranks for all the above 160 cases were calculated to estimate overall rank (right most column in Table 9) for an alternative. For example, if the alternative 1, maintenance of wall (see Table 9) has the highest probability for all

160 cases (4 building combo, 5 motivations, and 8 countries), the overall rank of alternative 1 is equal to 39.

The average ranks are also divided by country, building combo, and motivation. The average rank of an alternative for a country (say Spain) is the average values of ranks for 20 cases (4 building combo and 5 motivations). Similarly, the average rank of an alternative for a building combo (say R_SDB) is the average values of ranks for 40 cases (5 motivations and 8 countries). This average rank hypothesized to provide a reasonable metric to the relative importance given to a particular alternative in the choice-set ($K$). Therefore, the maximum average rank possible for an alternative is 39 (if an alternative has the highest probability in all cases) and minimum is 1 (if an alternative has the least probability in all cases). By identifying the highest average ranked alternatives for each particular case, we were able to validate the model, based on knowledge on the common choices in the residential building stock. Furthermore, it enables us to comprehend the extent to which each of these variables (building combo, motivation, and country) played a role in the selection of an alternative from the Choice set, as described in the sections below. The results were then validated and in line with the outcome of other researchers, as described in [20,43].

Table 10 shows the overview of rankings of the 39 alternatives as explained above. By the overall ranking, the top ten alternatives are (as highlighted in the table): Maintenance_Wall, Maintenance_Envelope + New _EnergyGeneration, Upgrade_Wall, Maintenance_Roof, Maintenance_Envelope + Upgrade_EnergyGeneration, New element_Energy Generation, New element_EnergyStorage, Upgrade_Roof, Upgrade_Envelope + New _EnergyGeneration, and New element_Ventilation. As expected, these EERMs address a single building element or technology, rather than multiple measures or elements. Most of them related to the building envelope. Additionally, most of them are related to maintenance and upgrade of the existing components. The table also shows variation of rankings across country, building combo, and motivation, the alternative 33 has the highest rank (35.24) in Spain, while alternative 17 is the highest in Belgium. In terms of building combo, the highest score in retrofit projects (both in SDBs and MDBs) is alternative 1 (31.34 and 34.03 respectively).

These top 10 alternatives by the overall rankings were further analysed by country, motivation, and building combo to observe how these vary across the various cases.

**Table 10.** Average ranking per alternative across countries, building combo, and motivation.

| # | Alternative Description | Country | | | | | | | | Building Combo | | | | Motivation | | | | | Overall Ranking |
|---|---|---|---|---|---|---|---|---|---|---|---|---|---|---|---|---|---|---|---|
| | | ES | IT | PL | DE | NL | UK | FR | BE | R_SDB | R_MDB | D_SDB | D_MDB | Env | Tech | Eco | Soc | Leg | |
| 1 | Maintenance_Wall | 30.40 | 25.80 | 29.04 | 28.80 | 29.72 | 29.40 | 30.60 | 31.08 | 31.38 | 34.03 | 31.88 | 30.20 | 34.08 | 30.88 | 35.48 | 21.08 | 34.63 | 30.50 |
| 2 | Maintenance_Windows | 17.20 | 19.20 | 13.88 | 24.16 | 24.08 | 23.28 | 25.76 | 34.56 | 26.50 | 15.08 | 27.68 | 26.53 | 29.78 | 17.70 | 20.33 | 25.65 | 25.33 | 23.33 |
| 3 | Maintenance_Roof | 4.60 | 25.72 | 31.08 | 29.76 | 20.72 | 30.52 | 31.84 | 36.64 | 26.03 | 29.35 | 27.05 | 34.93 | 30.58 | 27.53 | 25.55 | 28.90 | 28.23 | 27.59 |
| 4 | Maintenance_Basement | 28.36 | 21.68 | 17.36 | 26.16 | 11.72 | 15.36 | 15.76 | 18.88 | 10.60 | 27.68 | 24.25 | 22.98 | 20.95 | 19.23 | 21.95 | 22.28 | 19.43 | 20.27 |
| 5 | Upgrade_Wall | 29.92 | 24.76 | 29.24 | 28.20 | 20.12 | 23.12 | 30.04 | 29.60 | 28.23 | 32.45 | 28.98 | 27.83 | 31.20 | 27.45 | 24.30 | 28.53 | 30.35 | 27.90 |
| 6 | Upgrade_Windows | 27.80 | 9.36 | 25.80 | 14.92 | 25.32 | 23.56 | 26.92 | 35.00 | 27.50 | 16.95 | 28.80 | 27.38 | 26.88 | 25.23 | 30.20 | 17.40 | 25.33 | 24.37 |
| 7 | Upgrade_Roof | 30.80 | 25.00 | 29.16 | 15.96 | 28.44 | 23.52 | 29.04 | 20.08 | 26.28 | 30.68 | 27.58 | 26.03 | 29.18 | 26.30 | 30.88 | 31.38 | 15.70 | 26.23 |
| 8 | Upgrade_Basement | 4.08 | 23.00 | 26.32 | 25.68 | 27.76 | 15.56 | 29.12 | 22.24 | 24.45 | 19.85 | 25.35 | 23.88 | 24.70 | 22.40 | 17.48 | 27.28 | 22.90 | 22.47 |
| 9 | New element_Windows | 25.64 | 19.76 | 26.24 | 23.56 | 13.16 | 23.20 | 25.12 | 34.40 | 29.25 | 32.98 | 10.85 | 28.35 | 28.40 | 25.43 | 18.60 | 29.00 | 26.10 | 24.71 |
| 10 | New element_Roof | 6.44 | 18.76 | 25.12 | 11.12 | 25.04 | 29.40 | 26.12 | 34.60 | 25.70 | 29.25 | 12.83 | 25.00 | 20.20 | 24.13 | 27.08 | 19.95 | 24.58 | 22.66 |
| 11 | Maintenance_Ventilation | 1.08 | 13.96 | 8.00 | 17.20 | 12.28 | 15.56 | 18.76 | 12.72 | 20.35 | 8.60 | 8.68 | 10.28 | 9.28 | 18.05 | 15.30 | 9.03 | 13.45 | 12.50 |
| 12 | Upgrade_Ventilation | 14.40 | 22.08 | 27.24 | 13.48 | 14.00 | 17.08 | 19.24 | 22.36 | 5.95 | 27.53 | 23.63 | 23.03 | 19.68 | 17.43 | 25.15 | 17.38 | 18.00 | 19.27 |
| 13 | New element_Ventilation | 28.52 | 21.60 | 26.44 | 24.60 | 28.64 | 27.44 | 19.92 | 18.32 | 27.45 | 30.80 | 27.90 | 26.08 | 19.10 | 28.05 | 32.25 | 28.80 | 27.93 | 26.11 |
| 14 | Maintenance_EnergyGeneration | 30.08 | 22.56 | 28.20 | 26.80 | 13.60 | 26.24 | 14.36 | 26.80 | 27.28 | 30.83 | 17.53 | 26.40 | 19.20 | 28.73 | 27.78 | 24.93 | 24.28 | 24.45 |
| 15 | Upgrade_EnergyGeneration | 11.96 | 25.24 | 28.80 | 28.28 | 30.40 | 22.12 | 29.20 | 16.08 | 25.30 | 29.48 | 26.38 | 24.65 | 17.95 | 30.45 | 29.33 | 25.13 | 25.23 | 25.06 |
| 16 | New element_EnergyGeneration | 31.20 | 25.52 | 29.04 | 28.68 | 30.12 | 29.60 | 14.08 | 21.84 | 28.35 | 31.45 | 28.70 | 27.05 | 30.10 | 27.38 | 21.18 | 28.25 | 32.93 | 27.38 |
| 17 | New element_EnergyStorage | 14.28 | 26.20 | 14.36 | 27.80 | 30.72 | 25.00 | 30.12 | 36.76 | 26.48 | 30.65 | 27.38 | 26.13 | 21.05 | 30.50 | 30.13 | 26.33 | 26.30 | 26.48 |
| 18 | Maintenance_Heating | 19.92 | 12.80 | 4.28 | 7.24 | 16.40 | 15.16 | 18.72 | 22.60 | 8.85 | 5.95 | 27.48 | 13.05 | 16.50 | 14.58 | 17.28 | 15.80 | 15.50 | 14.83 |
| 19 | Upgrade_Heating | 12.60 | 6.60 | 5.32 | 18.16 | 11.12 | 8.92 | 4.44 | 22.12 | 13.03 | 9.65 | 7.95 | 3.83 | 12.23 | 11.23 | 13.20 | 12.03 | 11.28 | 10.80 |
| 20 | New elements_Heating | 26.28 | 17.24 | 12.64 | 3.56 | 22.68 | 23.24 | 14.24 | 8.12 | 22.38 | 13.73 | 22.40 | 9.33 | 17.95 | 16.10 | 19.48 | 17.63 | 16.80 | 16.69 |
| 21 | Maintenance_Cooling | 26.76 | 20.12 | 25.24 | 21.48 | 14.28 | 22.28 | 23.84 | 1.96 | 27.15 | 18.75 | 10.08 | 26.03 | 22.33 | 19.93 | 23.35 | 20.78 | 20.55 | 20.29 |
| 22 | Upgrade_Cooling | 11.48 | 6.48 | 16.12 | 10.00 | 2.32 | 6.92 | 6.24 | 22.20 | 8.70 | 5.13 | 4.25 | 11.28 | 11.48 | 10.50 | 11.78 | 10.83 | 10.38 | 9.77 |
| 23 | New elements_Cooling | 16.24 | 8.64 | 19.80 | 12.24 | 4.64 | 10.40 | 6.04 | 30.80 | 11.80 | 9.33 | 15.50 | 10.18 | 14.85 | 13.90 | 15.40 | 14.60 | 14.23 | 13.45 |
| 24 | Maintenance_CombHeatingCooling | 13.88 | 8.68 | 1.76 | 25.68 | 5.80 | 9.52 | 3.12 | 16.40 | 5.10 | 8.20 | 10.48 | 3.15 | 9.88 | 8.78 | 10.10 | 9.50 | 8.83 | 8.44 |
| 25 | Upgrade__CombHeatingCooling | 22.16 | 16.96 | 4.64 | 19.36 | 17.32 | 20.08 | 6.28 | 32.68 | 26.08 | 29.40 | 9.55 | 7.23 | 20.00 | 17.93 | 20.80 | 18.73 | 17.98 | 18.07 |
| 26 | New elements__CombHeatingCooling | 25.16 | 18.72 | 3.60 | 21.44 | 17.64 | 19.28 | 14.60 | 33.84 | 15.73 | 11.18 | 17.15 | 32.73 | 20.48 | 19.23 | 21.20 | 19.90 | 19.55 | 19.49 |
| 27 | Upgrade_Envelope | 28.48 | 24.04 | 33.28 | 16.72 | 28.12 | 16.88 | 33.84 | 12.04 | 25.95 | 19.05 | 27.53 | 26.03 | 26.08 | 23.93 | 23.15 | 28.15 | 19.58 | 24.28 |
| 28 | Upgrade_Envelope + Upgrade_Heating | 16.76 | 10.08 | 6.72 | 18.64 | 8.88 | 7.16 | 5.96 | 12.72 | 12.88 | 6.93 | 7.70 | 3.78 | 11.20 | 10.23 | 9.93 | 12.95 | 10.65 | 10.19 |
| 29 | Upgrade_Envelope + New _Heating | 31.08 | 23.20 | 14.60 | 3.72 | 20.04 | 19.48 | 16.64 | 2.72 | 21.48 | 11.45 | 21.83 | 9.75 | 16.95 | 15.10 | 15.78 | 19.00 | 16.45 | 16.43 |
| 30 | Maintenance_Envelope + Upgrade_Heating | 13.12 | 13.00 | 6.32 | 22.44 | 10.40 | 11.00 | 6.40 | 17.24 | 12.18 | 8.80 | 8.73 | 8.63 | 15.20 | 11.60 | 11.38 | 9.88 | 14.90 | 11.84 |
| 31 | Maintenance_Envelope + New _Heating | 26.12 | 28.24 | 13.88 | 7.08 | 23.20 | 27.00 | 17.88 | 4.84 | 19.65 | 13.10 | 22.33 | 19.13 | 21.93 | 17.70 | 17.55 | 14.70 | 21.75 | 18.59 |
| 32 | Upgrade_Envelope + Upgrade_EnergyGeneration | 15.24 | 28.60 | 33.16 | 27.56 | 26.44 | 16.80 | 34.52 | 7.20 | 24.23 | 24.45 | 25.50 | 24.53 | 17.18 | 28.48 | 22.85 | 27.15 | 24.15 | 24.00 |
| 33 | Upgrade_Envelope + New _EnergyGeneration | 35.24 | 30.48 | 33.60 | 29.32 | 27.00 | 24.16 | 16.12 | 11.28 | 26.83 | 26.55 | 27.75 | 26.08 | 27.68 | 24.93 | 17.60 | 29.50 | 31.03 | 26.18 |
| 34 | Maintenance_Envelope + Upgrade_EnergyGeneration | 12.88 | 34.64 | 31.68 | 35.20 | 30.48 | 27.44 | 34.16 | 11.36 | 24.53 | 27.83 | 27.15 | 33.05 | 26.58 | 30.95 | 25.83 | 22.28 | 30.98 | 27.47 |
| 35 | Maintenance_Envelope + New _EnergyGeneration | 30.64 | 31.80 | 32.16 | 33.72 | 28.44 | 30.80 | 19.92 | 16.88 | 24.73 | 29.15 | 27.98 | 35.83 | 33.08 | 28.23 | 20.85 | 24.18 | 34.78 | 28.42 |
| 36 | Maintenance_Heating system + Upgrade_EnergyGeneration | 5.92 | 12.56 | 4.36 | 6.88 | 16.28 | 10.68 | 17.36 | 4.04 | 5.03 | 3.58 | 19.95 | 8.58 | 7.05 | 12.93 | 11.95 | 10.35 | 10.45 | 9.88 |
| 37 | Maintenance_Energy generation + New _Heating | 23.80 | 17.40 | 12.28 | 3.64 | 9.76 | 18.60 | 6.72 | 2.04 | 17.95 | 12.28 | 10.73 | 8.03 | 9.15 | 14.80 | 13.83 | 12.78 | 12.33 | 12.12 |
| 38 | Maintenance_Envelope + Upgrade_EnergyGeneration | 12.72 | 10.84 | 32.84 | 28.52 | 26.28 | 15.48 | 22.76 | 6.96 | 19.65 | 21.68 | 20.73 | 20.70 | 23.68 | 18.20 | 11.80 | 26.08 | 18.65 | 19.86 |
| 39 | Maintenance_Envelope + New _EnergyGeneration | 16.76 | 28.68 | 16.40 | 27.64 | 26.64 | 18.76 | 34.20 | 28.00 | 25.35 | 25.28 | 26.45 | 25.95 | 19.75 | 28.33 | 23.75 | 27.73 | 25.15 | 24.99 |

### 4.2.1. Per Country

Figure 2 shows the ranking across countries of the overall ranking, the top ten alternatives. Results show that the probabilities could vary substantially across countries. For instance, Maintenance_Roof had a very low average in Spain but very high in Belgium (one of the highest in all of the results). Upgrade_Envelope + New _EnergyGeneration was very high in Spain and very low in France and Belgium. When comparing the overall scores across countries one can see that some countries did not vary significantly across the ten alternatives (e.g., Poland), whereas other countries varied significantly across the various measures (e.g., Belgium). This might indicate that some countries had more standardised approaches when it comes to upgrading the energy performance of the building, whereas other countries go for a more diversified approach with a wider range of measures. Additionally, the ten selected measures might be relevant for some countries but in others not all of these ten measures are relevant, only some of them. For example, the maintenance of the roof (alternative 3) might not be as relevant in Spain as in other countries given that there is not as much snow or rain as in other countries.

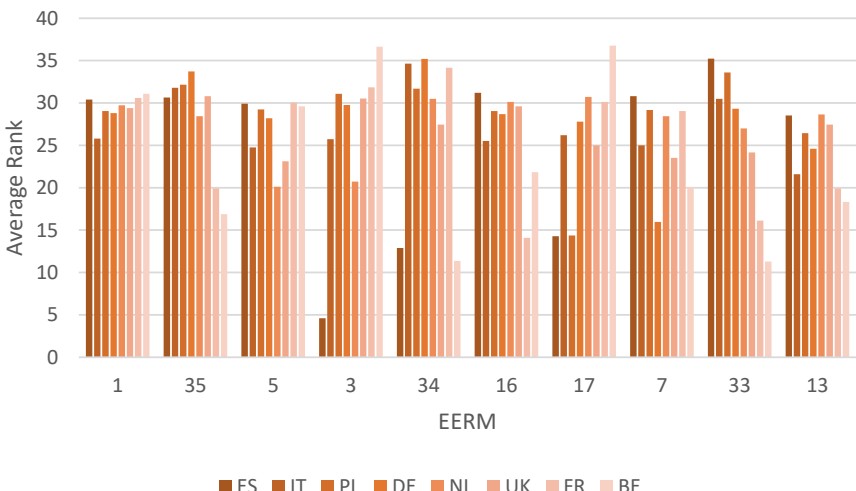

**Figure 2.** Ranking of top 10 alternatives, across countries. 1: Maintenance_Wall, 35: Maintenance_Envelope + New _EnergyGeneration, 5: Upgrade_Wall, 3: Maintenance_Roof, 34: Maintenance_Envelope + Upgrade_EnergyGeneration, 16: New element_EnergyGeneration, 17: New element_EnergyStorage, 7: Upgrade_Roof, Upgrade_Envelope + New _EnergyGeneration, 33: Upgrade_Envelope + New _Energy generation, 13: New element_Ventilation.

### 4.2.2. Per Building Combo

Figure 3 shows for the top ten alternatives, the ranking across building combo: retrofit of SDBs, retrofit of MDBs, deep retrofit of SDBs, and deep retrofit of MDBs. Results show that the average rankings are quite similar across all of these cases. These results could be explained on the basis that most of the EERMs are the same across building typologies and projects types. The main difference is then that in a deep retrofit project more EERMs are taking place than in a simple retrofit project. To better understand if this homogeneity was also present across countries, we further filtered the results per country. In this way we discovered that, for instance in Spain, the variation of ranks across building combo depended on the alternative that was analysed, for example, alternative 17 (New element_EnergyStorage) did not vary substantially across the building combo but alternative 18 (Maintenance_Heating) varied greatly.

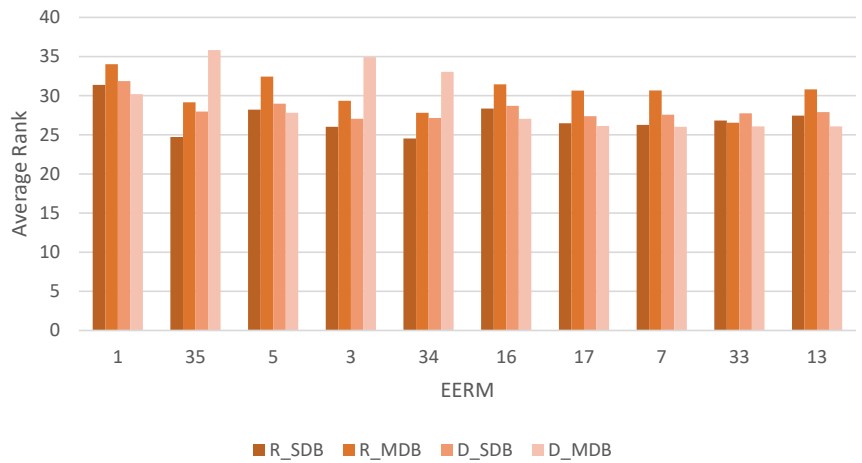

**Figure 3.** Ranking of alternative per building combo. 1: Maintenance_Wall, 35: Maintenance_Envelope + New _EnergyGeneration, 5: Upgrade_Wall, 3: Maintenance_Roof, 34: Maintenance_Envelope + Upgrade_EnergyGeneration, 16: New element_EnergyGeneration, 17: New element_EnergyStorage, 7: Upgrade_Roof, Upgrade_Envelope + New _EnergyGeneration, 33: Upgrade_Envelope + New _Energy generation, 13: New element_Ventilation.

Additionally, in all cases except for 35, 3, and 34, the highest average (rank) was the retrofit of MDBs. In these three exceptions it was the deep retrofit of MDBs the building combo with the highest average.

### 4.2.3. Per Motivation

Figure 4 presents the ten highest ranked alternatives across the motivations (i.e., environmental, technological, economic, social, and legal). Results show that the averages varied depending on the motivation behind the project. In most of these ten cases, the highest average was related to legal and environmental reasons, especially when it comes to alternative 1 (maintenance of the wall) and alternative 35 (Maintenance_Envelope + New _EnergyGeneration). In some cases, like in alternative 7 (Upgrade_Roof) and 33 (Upgrade_Envelope + New _EnergyGeneration), there was a great difference in ranking among the motivations, which shows that there was not an exclusive reason to implement these measures. The leading role of legal and environmental reasons behind these EERMs, is contrasting with some of the literature that argues that socioeconomic motivations are the most critical ones [23,44].

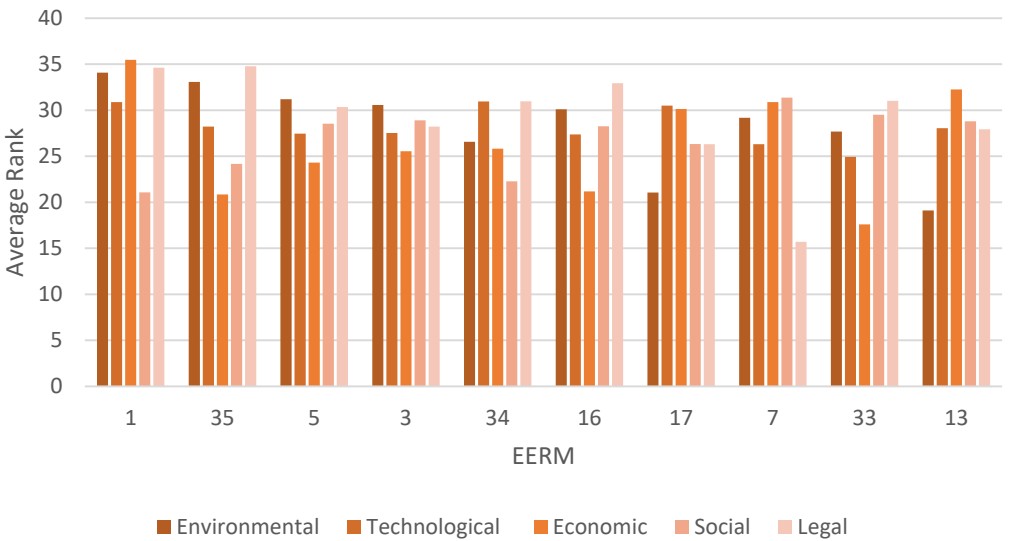

**Figure 4.** Ranking of alternative per Motivation. 1: Maintenance_Wall, 35: Maintenance_Envelope + New _EnergyGeneration, 5: Upgrade_Wall, 3: Maintenance_Roof, 34: Maintenance_Envelope + Upgrade_EnergyGeneration, 16: New element_EnergyGeneration, 17: New element_EnergyStorage, 7: Upgrade_Roof, Upgrade_Envelope + New _EnergyGeneration, 33: Upgrade_Envelope + New _Energy generation, 13: New element_Ventilation.

## 5. Discussion

In order to provide a cross-EU adoption of EERM in residential buildings, this study developed a modelling framework as a tool to estimate the probability of certain choice of EERMs to take place across eight EU countries (i.e., Spain, Italy, Poland, Germany, Netherlands, United Kingdom, France, and Belgium), according to a set of parameters, such as building typology (i.e., single- and multi-dwelling buildings), project types (i.e., simple or comprehensive retrofit), and motivation (i.e., social, economic, technical, and environmental). To the best of the authors' knowledge, this study is the first to provide an overview of EERMs across the selected variables and countries.

The methodology presented in this paper allows the comparison of EERMs across countries, motivations, and building combo, which is currently one of the biggest information flaws in the field of energy efficiency in the EU. A potential application of this model is in estimating market shares (provided the input data on the overall building projects in the country are available). This would, in turn, be useful information to evaluate the impact on certain policy measures aiming at fostering the uptake of energy efficiency measures. As it

could, for instance, showcase what happens to the EERM choices if there is a surge in the building combo 1 type of buildings by 10% or any subsidy or incentive for using building combo 1.

One of the limitations of the model is that to enable an accurate representation of the market, the survey questions required many answer options. This required the development of the logit model in order to collect all of the possible answer options.

In terms of the data, we collected a quota sample (not all of the projects in the market), which is not probabilistic. This also applies to the choice-set. It is assumed that the dataset used for estimating the DCMs is unbiased, although it was obtained from a stratified sample survey (not a random sample). Furthermore, we did not have the costs of each project or measure in the respective country, which limited the strength of the application of the DCM.

Additionally, the DCMs estimate the probabilities based on the choice-set, building combo, and the country. Since other potential variables such as building/establishment characteristics (floor area, business type, employment, etc.) and individual attributes (age, income, education, etc.) were not included; these models cannot capture the variation of the probabilities across these dimensions. For example, the probabilities estimated using these models, for two buildings (floor area 100 sqft and 1000 sqft, respectively) would be the same for a given choice-set, building combo, and country. Likewise, the high number of building variants (including exogenous and endogenous factors) can complicate the analysis and the interpretation of final results, as they depend on abundant calculation factors. Variables defining decision-makers were not included as the focus was to analyse how EERM varied across countries and not across decision-makers. Furthermore, there were already too many variables in place (thus computational time) but it would be contemplated for a further analysis.

Nonetheless, the modelling framework developed in this paper achieved the main purpose of the study that is to characterise cross-EU EERMs in residential buildings. While most of this paper's findings at a country level support the results from the previous studies described in the literature review [23,45,46], this work enables a consistent cross-country comparability of these observations. Furthermore, it provides additional insights into the building typologies and project types, and motivations behind these actions.

## 6. Conclusions

Results show that the impact of different countries (i.e., Spain, Italy, Poland, Denmark, Netherlands, UK, France, and Belgium), building combos (i.e., R_SDB, R_MDB, D_SDB, and D_MDB), and drivers (i.e., economic, environmental, technical, legal, and social) on the probability that certain measures or combinations of measures have been adopted in refurbishment and construction projects in the EU was highly significant. For instance, Netherlands was less likely to choose the maintenance of wall compared to UK and France. In terms of building combos, retrofit in single-dwelling buildings had higher probability of upgrading energy storage systems compared to that of deep retrofit single-dwelling buildings. Additionally, there were some similarities observed between a few drivers. For example, the economic and social drivers had similar probabilities of selecting a technology. Installation of any new technology was less likely to be chosen by legal and environmental drivers compared to that of other drivers. One of the key findings in respect to buildings was the demonstration that EERMs in residential buildings could vary across the eight EU sampled countries. EERMs that were identified as the top ten highest ranked were mostly related to the building envelope (i.e., Maintenance_Wall, Maintenance_Envelope + New _EnergyGeneration, Upgrade_Wall, Maintenance_Roof, Maintenance_Envelope + Upgrade_EnergyGeneration, New element_EnergyGeneration, New element_EnergyStorage, Upgrade_Roof, Upgrade_Envelope + New _EnergyGeneration, and New element_Ventilation). This finding can be attributed to the idea that actions to the building envelope addresses another relevant concern to building and dwelling owners, which is the aspect or appearance of the building and not necessarily cost-effective

EE actions. These EERMs related to the building envelope are complemented with other actions related to energy generation (New element_EnergyGeneration). As expected, these EERMs address a single building element or technology, rather than a cohort of these solutions. When looking at each of the choices, the intervention maintenance of the wall is high across all countries. The maintenance of the roof is high for all countries except Spain. The upgrade of the windows as a single intervention is high for all countries except for Italy and Germany. This can be partially attributed to policy instruments like "The Double Glazing Incentive" in Belgium [47]. The upgrade of the roof is high for all of countries except Germany and the UK. When it comes to including a new ventilation system, France and Belgium show the lowest values.

Findings suggest that the types of interventions that are currently undertaken are not necessarily correlating with those identified as most suitable to reach nZEB standards. Most of the adopted measures are addressing a single building element, out of which many are related to maintaining it, not necessarily upgrading the components to reach high energy efficiency standards. This is quite startling when we contrast it with the outcome of recent EU studies stating that 95% of the EU needs to be deeply retrofitted to meet carbon targets [48]. The modelling results also indicate that across all building typologies and project types, interventions to the building envelope are more likely to take place than heating and cooling systems. The intervention that has the lowest probability to take place is energy generation and energy storage.

The fact that the ranking averages vary substantially depending on the motivation behind the project also indicates that the motivations might be an important driver for the exact EERM that is undertaken. The level of influence depends on the specific EERM. Overall, the strongest motivation is identified as legal and environmental, which might indicate that legal and regulatory actions might be effective. Nevertheless, more research should be directed into trying to better understand to what extent these motivations determine the demand-side to take action in favour of a specific EERM in each market and building typology.

Some of the probabilities across countries also indicate that the EERMs that are currently being undertaken in building projects are not only driven to the energy performance of the building but also to construction practices. For instance, in Spain the maintenance of the roof has been historically less often addressed that in other countries such as Belgium or Germany. This might be due to the fact they there is less rain or snow throughout the year. Additionally, both in Belgium and Germany a number of policy instruments have been put place in order to incentivise this action [49]. This is portrayed by results such as that the Maintenance_Roof and has a very low average in Spain but very high in Belgium (one of the highest in all of the results) [50]. Another example of this is the upgrade of the basement.

Additionally, the selected combination of motivations is not entirely related to energy savings, e.g., social or legal. Thus, although the EERMs did address this matter, energy efficiency could not have necessarily been the primary intention of that measure, thus not the key motivation in the decision for adoption. Another key finding of the study is that, in most cases, the highest average is related to legal and environmental reasons. This supports some of the previous findings, such as [45] who identified that education and awareness on environmental matters could be propellers of residential energy-efficient technology adoption. This might be explained by the fact that most of the measures that were selected as the top ten had to do with the building envelope (e.g., maintenance of the wall and upgrade of the roof), and in many EU countries, the condition and maintenance of the façade is highly regulated in urban areas (e.g., France, Spain, and Belgium).

## 7. Outlook and Applications

In order to provide a comprehensive overview of the EU building sector, future research should focus on extending the findings of this study to the remaining EU countries, and non-residential buildings. Additionally, further data collection on EERMs related

to specific building typologies and geographical location should be collected. This can be supported by regulations and technologies enabling the monitoring of such measures under a cross-country consistent methodology to ensure the comparability of the results. Additionally, this information could be complemented with the specific rates of interventions in order to provide refurbishment rates across the EU broken down into the various building typologies and project types.

Likewise, advancing the understanding of the relationships between actors and how they are able to align interests, motivations, and EE actions could be a coherent next step in this research to better understand how and why these actions are taking place [46]. The data collected from the survey could help to guide where those gaps and opportunities exist in relation to EERMs.

A potential application of this study is as an evidence basis for the development of energy policy (national and pan-EU) with the aim of fostering the large-scale diffusion of energy-efficient technologies. In particular, by comparing the EERMs with the highest probability versus what should be happening in each country in order to achieve decarbonisation goals, these findings could support the identification of EERMs to be promoted in each country (or climatic zone), and across all member states.

**Author Contributions:** The corresponding author of this paper (C.C.) is responsible for initiating this paper, planning and conducting the research design and performing the analysis. The analysis and writing of the paper were done in close collaboration with L.K.K., I.H., I.S.-D. and L.R. supported in the conceptualization of the study. L.H. and K.B. contributed to the data collection. All authors have read and agreed to the published version of the manuscript.

**Funding:** This work was partly funded by the Danfoss Foundation and the Copenhagen Centre on Energy Efficiency (C2E2). The C2E2 is institutionally part of the UNEP DTU Partnership – operating under a tripartite agreement between the Danish Ministry of Foreign Affairs, UN Environment and the Technical University of Denmark (DTU). This work has also been partially financed by Climate-KIC, supported by the EIT – a body of the European Union TC_2.7.8_190515_P183-1B.

**Institutional Review Board Statement:** Not applicable.

**Informed Consent Statement:** Not applicable.

**Acknowledgments:** The authors would like to thank Zhuolun Chen, Gabriela Prata Dias and Merete Villum Pedersen. Special recognition goes to Martin Jakob (TEP Energy) and Paul Ruyssevelt at University College London and the EPSRC funded Centre for Research into Energy Demand Solutions [EP/R035288/1].

**Conflicts of Interest:** The authors declare no conflict of interest. The funders had no role in the design of the study; in the collection, analyses, or interpretation of data; in the writing of the manuscript, or in the decision to publish the results.

## Appendix A

*Appendix A.1. Survey Questionnaire and Quotas*

The first step in the questionnaire was to identify the respondent's role in the building value chain. To do this, respondents were asked: "Are you working professionally in one of the following companies or organisation types?" followed by a list of 21 options, including "Other company or organisation type in the building or construction sector" and "No, I do not work professionally in any company or organisation type related to the building and construction sector". To encompass all demand-side actors including private owners, for those interviewees who had indicated not to be working professionally in an organization from the building sector, there was the follow-up question "Do you privately own one or more residential home(s) or flat(s)?"

The "building combos" the EERM was embedded in, was collected by asking respondents "Within the last 3 years, have you completed a project in your country?". Subsequently, to know what exact measures had been implemented in their buildings, they were first asked to define what type of project they had last worked on, and then

provided with four answer options (or "building combos"): (a) overhaul or partial retrofit of single-dwelling buildings, (b) overhaul or partial retrofit of multi-dwelling buildings, (c) comprehensive retrofit project in single-dwelling buildings, and (d) comprehensive retrofit projects in multi-dwelling buildings. Respondents were then asked, "What measures were implemented in your latest project?" and given a table with 11 different elements covering all building components, which they had to choose from. Then they had to indicate what type of measure it was. The answer options were "Maintenance (including repair)", "Upgrade of existing elements or systems (incl. insulation and control)", and "New elements or systems". Additionally, they were provided the options of "I don't know" and "Other". The question allowed participants to choose more than one answer option.

In the next section (Section III), interviewees were asked: "What were the main motivations for your project?", and provided with 22 answer options, clustered into 5 categories: environmental, technical, economic, social, and legal, along with the choice "Other" and "I don't know". The format allowed participants to choose more than one answer option.

**Table A1.** Quotas.

| Stakeholder Group | Including (All These Stakeholder Groups Should Be Addressed) | New Built | | Deep Refurbishment (Three Elements or More) | | All Other Measures (Maintenance, Refurbishment) | | Sub-Total | | Grand-Total |
|---|---|---|---|---|---|---|---|---|---|---|
| | | SFH | MDH | SFH | MDH | SFH | MDH | SFH | MDH | |
| 1. Conceiving, planning, and consulting services | Architects and engineers | 9 | 9 | 9 | 9 | 9 | 9 | 27 | 27 | 54 |
| 2. Material and technology supply | Material or technology manufacturer or retailer | 9 | 9 | 9 | 9 | 9 | 9 | 27 | 27 | 54 |
| 3. Construction & installation | Construction companies and installers | 10 | 10 | 10 | 10 | 10 | 10 | 30 | 30 | 60 |
| 4. Enabling services | Local authorities, banks and other financial services | 9 | 9 | 9 | 9 | 9 | 9 | 27 | 27 | 54 |
| 5. Operation and maintenance services | Energy supply/utility and Energy service company (ESCO), facility managers: commercial, administrative, technical, maintenance, etc. | 10 | 10 | 10 | 10 | 10 | 10 | 30 | 30 | 60 |
| 6. Institutional demand side | Investors, developers, housing companies for profit, public/part Governmental/non-profit | 15 | 15 | 15 | 15 | 15 | 15 | 15 | 45 | 60 |
| 7. Private demand side | Private house owners, flats renter out or self-owned | 26 | 26 | 26 | 26 | 26 | 26 | 78 | 78 | 156 |
| Total | | 88 | 88 | 88 | 88 | 88 | 88 | 234 | 264 | 498 |

### Appendix A.2. Questionnaire

**Table A2.** Survey answer options to collect information of EERM implemented per building typology and project type.

| | Element or System | Maintenance (Including Repair) | Upgrade of Existing Elements or Systems (Incl. Insulation and Control) | New Element or Systems |
|---|---|---|---|---|
| | | A | B | C |
| E1 | Wall (outer) | ☐ | ☐ | ☐ |
| E2 | Windows | ☐ | ☐ | ☐ |
| E3 | Roof (pitched/flat) or attic | ☐ | ☐ | ☐ |
| E4 | Basement/crawl space | ☐ | ☐ | ☐ |
| T1 | Ventilation system | ☐ | ☐ | ☐ |
| T2 | Energy generation (PV or solar collector) | ☐ | ☐ | ☐ |
| T3 | Energy storage | ☐ | ☐ | ☐ |
| T4 | Appliances Please specify if possible: | ☐ | ☐ | ☐ |
| T5 | Heating system | ☐ | ☐ | ☐ |
| T6 | Cooling system Please specify what type of cooling system, if possible | ☐ | ☐ | ☐ |
| T7 | Combined heating & cooling system (reversible heat pump) | ☐ | ☐ | ☐ |
| O1 | Other: Please describe | ☐ | ☐ | ☐ |

### Appendix A.3. EERM Choice-Set

The choice-set used in this study was based on the findings and it comprised of a comprehensive list of 39 EERMs. As some of these interventions, the 39 EERMs can be broken down into 63 possibilities (see "Reference" column in Table A3). The combinations were developed based on the most cost-effective combinations to reach nZEB and cost-optimal for different building types and EU climatic zones according to (Zangheri et al., 2016). The list was then validated through discussions with market experts for each country. The complete list of discrete choices used in the numerical analysis can be found in the Table A3 below.

**Table A3.** Building interventions (BIs) used as discrete alternatives in the model estimation.

| # Alternative | Definition | Reference (see Table A2) | Number of Unique Combinations of EERMs |
|---|---|---|---|
| 1 | Maintenance_Wall | A-E1 | 1 |
| 2 | Maintenance_Windows | A-E2 | 1 |
| 3 | Maintenance_Roof | A-E3 | 1 |
| 4 | Maintenance_Basement | A-E4 | 1 |
| 5 | Upgrade_Wall | B-E1 | 1 |
| 6 | Upgrade_Windows | B-E2 | 1 |
| 7 | Upgrade_Roof | B-E3 | 1 |
| 8 | Upgrade_Basement | B-E4 | 1 |
| 9 | New element_Windows | C-E2 | 1 |
| 10 | New element_Roof | C-E3 | 1 |

**Table A3.** *Cont.*

| #<br>Alternative | Definition | Reference<br>(see Table A2) | Number of Unique<br>Combinations of<br>EERMs |
|---|---|---|---|
| 11 | Maintenance_Ventilation | A-T1 | 1 |
| 12 | Upgrade_Ventilation | B-T1 | 1 |
| 13 | New element_Ventilation | C-T1 | 1 |
| 14 | Maintenance_Energy generation (e.g., PV) | A-T2 | 1 |
| 15 | Upgrade_Energy generation (e.g., PV) | B-T2 | 1 |
| 16 | New element_Energy generation (e.g., PV) | C-T2 | 1 |
| 17 | New element_Energy storage | C-T3 | 1 |
| 18 | Maintenance_Heating system | A-T5 | 1 |
| 19 | Upgrade_Heating system | B-T5 | 1 |
| 20 | New elements_Heating system | C-T5 | 1 |
| 21 | Maintenance_Cooling system | A-T6 | 1 |
| 22 | Upgrade_Cooling system | B-T6 | 1 |
| 23 | New elements_Cooling system | C-T6 | 1 |
| 24 | Maintenance_Combined heating and cooling | A-T7 | 1 |
| 25 | Upgrade_Combined heating and cooling | B-T7 | 1 |
| 26 | New elements_Combined heating and cooling | C-T7 | 1 |
| 27 | Upgrade_Envelope | B(E1 + E2 + E3 + E4), B(E1 + E2 + E3), B(E1 + E2 + E4), B(E1 + E3 + E4), B(E2 + E3 + E4) | 5 |
| 28 | Upgrade_Envelope + Upgrade_Heating system | B(E1 +E2) + B-T5, B(E1 +E3) + B-T5, B(E1 +E4) + B-T5 | 3 |
| 29 | Upgrade_Envelope + New _Heating system | B(E1 +E2) + C-T5, B(E1 +E3) + C-T5, B(E1 +E4) + C-T5 | 3 |
| 30 | Maintenance_Envelope + Upgrade_Heating system | A(E1 + E2) + B-T5, A(E1 + E3) + B-T5, A(E1 + E4) + B-T5 | 3 |
| 31 | Maintenance_Envelope + New _Heating system | A(E1 + E2) + C-T5, A(E1 + E3) + C-T5, A(E1 + E4) + C-T5 | 3 |
| 32 | Upgrade_Envelope + Upgrade_Energy generation | B(E1 + E2)) + B-T2, B(E1 + E3)) + B-T2, B(E1 + E4)) + B-T2 | 3 |
| 33 | Upgrade_Envelope + New _Energy generation | B(E1 +E2) + C-T2, B(E1 +E3) + C-T2, B(E1 +E4) + C-T2 | 3 |
| 34 | Maintenance_Envelope + Upgrade_Energy generation | A(E1 + E2) + B-T2, A(E1 + E3) + B-T2, A(E1 + E4) + B-T2 | 3 |
| 35 | Maintenance_Envelope + New _Energy generation | A(E1 +E2) + C-T2, A(E1 +E3) + C-T2, A(E1 +E4) + C-T2 | 3 |
| 36 | Maintenance_Heating system + Upgrade_Energy generation | A-T5 + B-T2 | 1 |
| 37 | Maintenance_Energy generation + New _Heating system | A-T2 + C-T5 | 1 |
| 38 | Upgrade_Envelope + Upgrade_Energy storage | B(E1 + E2)) + B-T3, B(E1 + E3)) + B-T3, B(E1 + E4)) + B-T3 | 3 |
| 39 | Upgrade_Envelope + New _Energy storage | B(E1 +E2) + C-T3, B(E1 +E3) + C-T3, B(E1 +E4) + C-T3 | 3 |
| | Total | | 63 |

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
