# Peer review of "Energy-Efficient Retrofit Measures (EERM) in Residential Buildings: An Application of Discrete Choice Modelling"

_buildings, doi:10.3390/buildings11060257_

Round 1
Reviewer 1 Report
This is an interesting study indeed. The subject is of great interest to a wide range of stakeholders and the methodology used has transferable potential to other disciplines. The language very clear and to the point.
I would like to see the two following points addressed before this paper is accepted for publication (this may include discussing further uncertainties or providing more clarifications- I am not suggesting a major revision here):
- There is no reference to the language(s) used in the survey. Was the surveyed administered in local/native language in each country or done in English? If the latter, the implications of varying language skills needs to be discussed. If the former, the protocol for translations needs to be discussed.
- There are two occasions where input from experts is discussed, implying also that this has largely influenced the design of critical aspects of the study. The reader needs to be provided with more detail on how such consultations were performed. Especially with regards to the development of the questionnaire- it is unusual for the research team not to take the lead on the way questions and answer options are worded. This needs to be discussed in some detail.
- There are a number of schemes that may have influenced the results observed. See http://www.buildup.eu/en/nationalregional-schemes-individuals-homeowners-tenants?page=7 for breakdown per country (note UK data is about to be withdrawn from this list). The research team needs to relate these results to the schemes that have incentivise these upgrades. In many cases owners or tenants are able to make own choices out of options provided, so the motivation variable would still be relevant here. It just looks odd that the upgrading of windows is not listed higher- to what extent is this because for some participants this type of upgrade would be described as a glazing replacement or using other combinations of words?

Author Response
This is an interesting study indeed. The subject is of great interest to a wide range of stakeholders and the methodology used has transferable potential to other disciplines. The language very clear and to the point.
I would like to see the two following points addressed before this paper is accepted for publication (this may include discussing further uncertainties or providing more clarifications- I am not suggesting a major revision here):
- There is no reference to the language(s) used in the survey. Was the surveyed administered in local/native language in each country or done in English? If the latter, the implications of varying language skills needs to be discussed. If the former, the protocol for translations needs to be discussed
- The survey was translated to the official language of each country.
We have added an explanation to address this point. Lines 169-171:
“The final questionnaire was then translated into the language and jargon of each country. The translations were revised by market experts in each country to ensure the correct understanding and interpretation of the questions within their context.”
- There are two occasions where input from experts is discussed, implying also that this has largely influenced the design of critical aspects of the study. The reader needs to be provided with more detail on how such consultations were performed. Especially with regards to the development of the questionnaire- it is unusual for the research team not to take the lead on the way questions and answer options are worded. This needs to be discussed in some detail.
- We have included a paragraph to address this aspect. Lines 264-272:
“To test the validity of the survey design and examine any potential flaws in the research conception, a workshop was organized gathering stakeholder representatives from all relevant groups in the building value chain (i.e. supply-side actors, demand-side actors and enablers). More than 20 participants attended the workshop. The input received validated the survey design and served as a valuable basis for drafting the questionnaire.”
- There are a number of schemes that may have influenced the results observed. See http://www.buildup.eu/en/nationalregional-schemes-individuals-homeowners-tenants?page=7 for breakdown per country (note UK data is about to be withdrawn from this list). The research team needs to relate these results to the schemes that have incentivise these upgrades. In many cases owners or tenants are able to make own choices out of options provided, so the motivation variable would still be relevant here. It just looks odd that the upgrading of windows is not listed higher- to what extent is this because for some participants this type of upgrade would be described as a glazing replacement or using other combinations of words?
- Thank you for sharing the reference. This is a very relevant point that we now have addressed in the “2. Theoretical framework” as well as “6. Conclusions” sections.
Line 58:
“Given the diverse national organization that composes the EU, it is particularly important to identify country-specific differences in energy-saving technology adoption patterns to generate an appropriate combination of common and country-specific policies [17,18]”
Line 189-190:
“This can be partially attributed to policy instruments like “The Double Glazing Incentive” in Belgium [46].”
Line 207-208:
Also, both in Belgium and Germany a number of policy instruments have been put place in order to incentivise this action [48].

Reviewer 2 Report
- In the introduction, among the strategies of energy improvement of the building heritage, a particular mention to the Life Cycle Cost analysis (LCC) could be made (e.g. doi: 10.3390/su11051452, doi: 10.3390/en10111851), as a consolidated tool to guide choices at the level of the single building but also at the national policy level.
- A comment could be useful to further clarify the "conditionally logit (CL) model based on the random utility theory" used by the authors in the analysis.
- Is the model indicated in the eq.(6) (different from those of eqs. (2)-(5) to capture the effect of country, building combo, and motivation simultaneously) still based on linear relations between the parameters? A comment would be useful and necessary.
- There is an error (reference source) at line 406.
- The conclusions seem somewhat lacking in quantitative data.
Author Response
- In the introduction, among the strategies of energy improvement of the building heritage, a particular mention to the Life Cycle Cost analysis (LCC) could be made (e.g. doi: 10.3390/su11051452, doi: 10.3390/en10111851), as a consolidated tool to guide choices at the level of the single building but also at the national policy level.
- Thank you for this comment. We have included this in the manuscript now. See lines 104-108:
“As a consolidated tool to guide choices at the level of the single building as well as national policy level, R. Moschetti et al. combined life-cycle environmental and economic assessments in building energy renovation project of a single-family house in Norway. The results demonstrated the close to negative linear regression between the environmental and economic indicators that were computed [26].”
- A comment could be useful to further clarify the "conditionally logit (CL) model based on the random utility theory" used by the authors in the analysis.
- Thank you for the suggestion. We included the following text in the Section 3.3 Modelling approach to further clarify the random utility theory and updated the Equation (7) accordingly. See lines 363 to 369.
“The RUT is based on the hypothesis that the users (choice-makers) are rational individuals who try to maximize the perceived utility from the choice made. Hence, the probability that an alternative k is selected is equal to the probability that the utility derived by the user in choosing k is higher than utilities of choosing any other alternatives apart from k (see (7)). The utilities could be measured as a linear function of attributes that are either relevant to the choice or the user. In this research, the utilities of making different choices of EERMs are estimated as a function of country, building combo and motivation behind the project as explained in (2) to (6).”
- Is the model indicated in the eq.(6) (different from those of eqs. (2)-(5) to capture the effect of country, building combo, and motivation simultaneously) still based on linear relations between the parameters? A comment would be useful and necessary.
- Thank you for pointing this. We have included the following lines in the Section 3.3.2 Model Formulations: Combined all main effects. Please see lines 432 to 437.
“For example, (3) could be used to compare the probabilities of selecting an EERM across different countries but cannot be used to study the same across different building combos. Similarly, (4) helps in studying the choice of EERMs exclusively across building combos. Whereas, (6) could quantify the choice of EERMs at the disaggregated level such as what is the probability of an EERM at a given country, building combo and motivation of the project e.g., in Spain (ES) belonging to the retrofit of Single-dwelling building (R_SDB), with an Environmental (Env) motivation?
- There is an error (reference source) at line 406
- Thank you for letting us know. The reference source has been updated accordingly.
- The conclusions seem somewhat lacking in quantitative data.
- Thank you for this comment. We very much agree. We have added the following quantitative information in the conclusions section (Lines 166 and 175):
“Results show that the impact of different countries (Spain, Italy, Poland, Denmark, Netherlands, UK, France, and Belgium), building combos (R_SDB, R_MDB, D_SDB, D_MDB), and drivers (i.e. economic, environmental, technical, legal, social) on the probability that certain measures or combinations of measures have been adopted in refurbishment and construction projects in the EU is highly significant. For instance, Netherlands is less likely to choose the maintenance of wall compared to UK and France. In terms of building combos, retrofit in single-dwelling buildings have higher probability of upgrading energy storage systems compared to that of deep retrofit single-dwelling buildings. Also, there are some similarities observed between a few drivers. For example, the economic and social drivers have similar probabilities of selecting a technology. Installation of any new technology is less likely to be chosen by legal and environmental drivers compared to that of other drivers.”

Reviewer 3 Report
Dear authors,
Thank you for your manuscript, which I read with great interest. However, I think some opportunities exist to further improve it.
Whole paper
When referring to authors in person, often only the family name is used without the first letter. In some cases the authors use only the family name, but in most cases also the first letter (e.g. line 36, 79, 82, 87, 90, 93 etc).
Please try to be slightly more concise on the research methodology and model development. The readers of Buildings might appreciate when you focus more on what your research results mean to "buildings" and "energy policies".
The section and page numbers are not well established in the pdf-file. In general try to focus on what is relevant for the readers of buildings, who aren't all of always model developers. Most of our readers want to understand and to improve the built environment... so conciseness on research method and model building would be appreciated. Please, help the reader to interpret what your findings mean in adopting energy techniques and measures in buildings.
Now some more specific points of attention will follow. Please feel free to use it to your advantage.
Abstract
r. 13 building elements
"This study contributes to the scientific community 14 by developing a modelling framework to establish an empirical relationship among EERMs and 15 project (i.e. retrofit and deep retrofit) and building characteristics across these countries." This sentence contains the word "and" two times which makes it unclear to the reader what is related to what exactly? 3 items; 6 relations?
1. Introduction
Maybe it would improve the readability to distinguish an introduction and a 2. Theory / 2. Theoretical background / 2. Theoretical framework section, before line 73.
2. Materials and Methods
Maybe the authors can consider to rename this section; 3. Research methodology.
Line 175 this step worries me slightly, because how was the questionnaire distributed exactly? Who received the questionnaires how?
The second Section 3.1.1; this section might be just a first appendix, to which you can refer in section 3.
Line 240 Why is the sample size adequate? Because you went from around 500 to only 50 in one case? What is the basis of this statement?
The section that is numbered as section 2 and 3 needs some restructuring and conciseness.
Line 406 a reference went lost.
Section 3.3.3 please explain to the reader how the elasticity needs to be interpreted in relations to the context of energy techniques and measures?
4. Model Results
Line 450 what black brackets exactly?
Figure 1, 2 and 3: number 33 was not explained. My suggestion would be to apply the explaining text on the numbers directly beneath the axis in the figure.
Line 85 The results could be interpreted by the fact... Sorry, but I do not completely understand what you are aiming at here?
Line 111-113 Please be specific what literature you are referring to here exactly? You haven't got cost data, so getting into economics might be tricky here. Where would you like to come back to your own literature framework as a basis to discuss existing insights in relation to your new insights?
Conclusions
In the conclusions I would like to see clearer straight forward answers to the research questions raised in Section 1 Introduction.
Best regards,
a reviewer
Author Response
Dear authors,
Thank you for your manuscript, which I read with great interest. However, I think some opportunities exist to further improve it.
Whole paper
When referring to authors in person, often only the family name is used without the first letter. In some cases the authors use only the family name, but in most cases also the first letter (e.g. line 36, 79, 82, 87, 90, 93 etc)
- Thank you for pointing this out. We have gone through the text again to harmonise the referral to the authors including only the family name.
Please try to be slightly more concise on the research methodology and model development.
- Thank you for this suggestion. We have further worked on the methodology section to make it more concise, particularly in those sections where the other reviewers had asked for further details. Please find it in lines: 160-162, 155-159 and 195-199.
The readers of Buildings might appreciate when you focus more on what your research results mean to "buildings" and "energy policies”.
- Thank you for the comment. We included the below text in the Section 6. Conclusions to explain how our research could be helpful in estimating the outcomes with respect to energy policies.
Lines 166-175:
“Results show that the impact of different countries (Spain, Italy, Poland, Denmark, Netherlands, UK, France, and Belgium), building combos (R_SDB, R_MDB, D_SDB, D_MDB), and drivers (i.e. economic, environmental, technical, legal, social) on the probability that certain measures or combinations of measures have been adopted in refurbishment and construction projects in the EU is highly significant. For instance, Netherlands is less likely to choose the maintenance of wall compared to UK and France. In terms of building combos, retrofit in single-dwelling buildings have higher probability of upgrading energy storage systems compared to that of deep retrofit single-dwelling buildings. Also, there are some similarities observed between a few drivers. For example, the economic and social drivers have similar probabilities of selecting a technology. Installation of any new technology is less likely to be chosen by legal and environmental drivers compared to that of other drivers.”
Lines 135-136:
A potential application of this study is as an evidence basis for the development of energy policy (national and pan-EU) with the aim of fostering the large-scale diffusion of energy-efficient technologies.
The section and page numbers are not well established in the pdf-file. In general try to focus on what is relevant for the readers of buildings, who aren't all of always model developers. Most of our readers want to understand and to improve the built environment... so conciseness on research method and model building would be appreciated. Please, help the reader to interpret what your findings mean in adopting energy techniques and measures in buildings.
- Thank you for highlighting this very relevant point. We have revised the whole manuscript in order to help the reader interpret our findings as well as possible. Amendments have been implemented across all sections with particular focus on the methods and conclusions sections.
Now some more specific points of attention will follow. Please feel free to use it to your advantage.
Abstract
- 13 building elements
- Thank you for identifying the typo. We have corrected it accordingly.
"This study contributes to the scientific community 14 by developing a modelling framework to establish an empirical relationship among EERMs and 15 project (i.e. retrofit and deep retrofit) and building characteristics across these countries." This sentence contains the word "and" two times which makes it unclear to the reader what is related to what exactly? 3 items; 6 relations?
- Thank you this comment. Indeed, it was not clear. We have rephrased it by the following text. See lines 23-26: “The modelling framework developed in this study contributes to the scientific community in two three ways: (1) establishing an empirical relationship among EERMs and project (i.e. retrofit and deep retrofit), and (2) identifying commonalities and differences across the selected countries, and (3) quantifying the probabilities and market shares of various EERMs.”
- Introduction
Maybe it would improve the readability to distinguish an introduction and a 2. Theory / 2. Theoretical background / 2. Theoretical framework section, before line 73.
- We very much agree with this point. We have added the section “2. Theoretical framework” before (former) line 73 accordingly. The numbering of the remaining sections are also updated on that basis.
- Materials and Methods
Maybe the authors can consider to rename this section; 3. Research methodology.
- We have changed the name of the section to “3. Research methodology”
Line 175 this step worries me slightly, because how was the questionnaire distributed exactly? Who received the questionnaires how?
- The survey distribution was conducted following a rigorous procedure based on the statistical requirements. The analysis required for a random sample selected for each country and so it was executed. We have added further explanation in the manuscript to describe this. Please see lines 199-203: “To obtain a well-balanced sample, the distribution of the survey was assigned to a professional company with a pan-European presence, which was able to follow a consistent methodology across all countries. The distribution collected a random sample. Since the analysis focused on building projects, a subset of the complete database was used, excluding any response which did not contain a project type.”
The second Section 3.1.1; this section might be just a first appendix, to which you can refer in section 3.
- We have moved Section 3.1.1 as Appendix 1 and referred to it in section 3.
Line 240 Why is the sample size adequate? Because you went from around 500 to only 50 in one case? What is the basis of this statement?
- Thank you for the suggestions. We added the below text to explain better. See lines 265 to 267: “Based on the previous studies on the discrete choice models, the sample size required to capture the choice should be at least 35. Hence, the dataset used in this study is adequate for the discrete choice modelling.”
The section that is numbered as section 2 and 3 needs some restructuring and conciseness –
- Thank you for this suggestion. We have further worked on the methodology section to make it more concise, particularly in those sections where the other reviewers had asked for further details. Please find it in lines: 164-172, 176, 199-203, 265-267, 363-369, 432-437, 459-462, and 491-492.
Line 406 a reference went lost
- Thank you for letting us know. The reference source has been updated accordingly.
Section 3.3.3 please explain to the reader how the elasticity needs to be interpreted in relations to the context of energy techniques and measures?
- This is a very valuable suggestion. We appreciate it. The interpretation of elasticities is included in Section 3.3.3 Model Formulations: Probability and Elasticities. See lines 459-462: “In this context, the elasticities in (8) and (9) estimate the relative change in the probability of selecting an EERM in a country, building combo, and motivation of the project if an EERM is included or excluded in the choicest. For example, what would be percentage increase in the probability of selecting the maintenance of wall if installing new wall is not a possibility anymore?”
- Model Results
Line 450 what black brackets exactly?
- We apologize for the typo-related confusion. We changed the text to clarify this. See lines 463-464: “The blank cells in the Table 8.2 indicate the parameters which are not significantly different from zero.”
Figure 1, 2 and 3: number 33 was not explained. My suggestion would be to apply the explaining text on the numbers directly beneath the axis in the figure
- Thank you for pointing this out. We included the explanation for number 33 below the figures 1, 2 and 3. We tried including the explaining text in the axis labels, which made the figure look clumsy with a weird looking X-axis. Hence, for aesthetic purposes, we stick to including the explanation below the figure. We apologize for this.
Line 85 The results could be interpreted by the fact... Sorry, but I do not completely understand what you are aiming at here?
- Thank you for highlighting this issue. We have revised the sentence accordingly. Please see line 86: “These results could be explained on the basis that most”
Line 111-113 Please be specific what literature you are referring to here exactly? You haven't got cost data, so getting into economics might be tricky here. Where would you like to come back to your own literature framework as a basis to discuss existing insights in relation to your new insights? –
- Thank you for this comment. We have added a clarification of this references in lines 111-114: “which argues that it is socio-economic motivations are the most critical ones [26,44].”
Conclusions
In the conclusions I would like to see clearer straight forward answers to the research questions raised in Section 1 Introduction _
- Thank you for this comment. To strengthen the link between the research questions and the conclusions, we have included two paragraphs at the beginning of the conclusions section explicitly addressing this. See lines 165-174: “Results show that the impact of different countries (Spain, Italy, Poland, Denmark, Netherlands, UK, France, and Belgium), building combos (R_SDB, R_MDB, D_SDB, D_MDB), and drivers (i.e. economic, environmental, technical, legal, social) on the probability that certain measures or combinations of measures have been adopted in refurbishment and construction projects in the EU is highly significant. For instance, Netherlands is less likely to choose the maintenance of wall compared to UK and France. In terms of building combos, retrofit in single-dwelling buildings have higher probability of upgrading energy storage systems compared to that of deep retrofit single-dwelling buildings. Also, there are some similarities observed between a few drivers. For example, the economic and social drivers have similar probabilities of selecting a technology. Installation of any new technology is less likely to be chosen by legal and environmental drivers compared to that of other drivers.”
Best regards,
a reviewer

Round 2
Reviewer 3 Report
Dear authors,
Please be more specific in line 259 "Based on the previous studies on the discrete choice 259 models, the sample size required to capture the choice should be at least 35. Hence, the dataset used 260 in this study is adequate for the discrete choice modelling.." Here I would like you to back these specific previous up with multiple valuable references in this area of research, please.
Line 112 "The leading role of legal and environmental reasons behind these EERMs, 112 is contrasting with some of the literature that argues that economic reasons are the most 113 relevant, which argues that it is socio-economic motivations are the most critical ones 114 [26,44]." This sentence needs a revision I am afraid, because two times argues doesn't make it very clear now.
The conclusion section is still relatively vaguely formulated; what are exactly the answers to your two research questions: • What is the impact of different drivers (i.e. economic, environmental, technical, legal, social) 62 on the probability that certain measures or combinations of measures have been adopted in 63 refurbishment and construction projects in the EU? 64
• How does this probability of technology adoption differ across building envelope measures, 65 heating and cooling systems and appliances?
Maybe the answer to your second question is already provided a section earlier. So you might want to focus on formulating one 'how' research question that you clearly provide an answer to in the conclusion section.
Adjustments already made are appreciated and really improved the paper.
Best regards,
A reviewer
Author Response
- Please be more specific in line 259 "Based on the previous studies on the discrete choice 259 models, the sample size required to capture the choice should be at least 35. Hence, the dataset used 260 in this study is adequate for the discrete choice modelling.." Here I would like you to back these specific previous up with multiple valuable references in this area of research, please.
- Thank you for the valuable suggestion. We have included the following references:
- Ben-Akiva, M., Mcfadden, D., Abe, M. et al. "Modeling Methods for Discrete Choice Analysis". Marketing Letters 8, 273–286 (1997). https://doi.org/10.1023/A:1007956429024
- Chau, Chi Kwan, M. S. Tse, and K. Y. Chung. "A choice experiment to estimate the effect of green experience on preferences and willingness-to-pay for green building attributes." Building and Environment 45, no. 11 (2010): 2553-2561.
- Michelsen, Carl Christian, and Reinhard Madlener. "Homeowners' preferences for adopting innovative residential heating systems: A discrete choice analysis for Germany." Energy Economics 34.5 (2012): 1271-1283.
- Wang, Jingjie, Hongbin Wu, Shihai Yang, Rui Bi, and Junhua Lu. "Analysis of decision-making for air conditioning users based on the discrete choice model." International Journal of Electrical Power & Energy Systems 131 (2021): 106963.
- Thank you for the valuable suggestion. We have included the following references:
- Line 112 "The leading role of legal and environmental reasons behind these EERMs, is contrasting with some of the literature that argues that economic reasons are the most relevant, which argues that it is socio-economic motivations are the most critical ones 114 [26,44]." This sentence needs a revision I am afraid, because two times argues doesn't make it very clear now.
- Thank you for highlighting this sentence. Indeed, it needed revision. We have exchanged it by “The leading role of legal and environmental reasons behind these EERMs, is contrasting with some of the literature that argues that socio-economic motivations are the most critical ones [26,44].”
- The conclusion section is still relatively vaguely formulated; what are exactly the answers to your two research questions:
- What is the impact of different drivers (i.e. economic, environmental, technical, legal, social) on the probability that certain measures or combinations of measures have been adopted in refurbishment and construction projects in the EU?
- Thank you for this remark. We have added the following information to the conclusions section to answer to this question:
- What is the impact of different drivers (i.e. economic, environmental, technical, legal, social) on the probability that certain measures or combinations of measures have been adopted in refurbishment and construction projects in the EU?
“The level of influence depends on the specific EERM. In overall, the strongest motivation is identified as legal and environmental, which might indicate that legal and regulatory actions might be effective. Nevertheless, more research should be directed into trying to better understand to what extent do these motivations determine the demand-side to take action in favour of a specific EERM in each market and building typology.
- How does this probability of technology adoption differ across building envelope measures, heating and cooling systems and appliances?
- Thank you for defining this question. We have added additional information in the conclusions section to address it:“The modelling results also indicate that across all building typologies and project types, interventions to the building envelope are more likely to take place than heating and cooling systems. The intervention that has the lowest probability to take place is energy generation and energy storage.”